# Adversarial Robustness in Two-Stage Learning-to-Defer: Algorithms and Guarantees

**Yannis Montreuil** [1 2 3]   **Axel Carlier** [4 5]   **Lai Xing Ng** [2 5]   **Wei Tsang Ooi** [1 5]

## Abstract

Two-stage Learning-to-Defer (L2D) enables optimal task delegation by assigning each input to either a fixed main model or one of several offline experts, supporting reliable decision-making in complex, multi-agent environments. However, existing L2D frameworks assume clean inputs and are vulnerable to adversarial perturbations that can manipulate query allocation—causing costly misrouting or expert overload. We present the first comprehensive study of adversarial robustness in two-stage L2D systems. We introduce two novel attack strategies—*untargeted* and *targeted*—which respectively disrupt optimal allocations or force queries to specific agents. To defend against such threats, we propose SARD, a convex learning algorithm built on a family of surrogate losses that are provably Bayes-consistent and $(\mathcal{R}, \mathcal{G})$-consistent. These guarantees hold across classification, regression, and multi-task settings. Empirical results demonstrate that SARD significantly improves robustness under adversarial attacks while maintaining strong clean performance, marking a critical step toward secure and trustworthy L2D deployment.

## 1. Introduction

Learning-to-Defer (L2D) is a powerful framework that enables decision-making systems to optimally allocate queries among multiple agents, such as AI models, human experts, or other decision-makers (Madras et al., 2018). In the *two-stage* framework, agents are trained offline to harness their domain-specific expertise, enabling the allocation of each task to the decision-maker with the highest confidence (Mao et al., 2023a; 2024d; Mao, 2025; Montreuil et al., 2024; 2025b). L2D is particularly valuable in high-stakes applications where reliability and performance are critical (Mozannar & Sontag, 2020; Verma et al., 2023). In healthcare, L2D systems integrate an AI diagnostic model with human specialists, delegating routine tasks to AI and deferring edge cases—such as difficult images—to specialists for nuanced evaluation (Mozannar et al., 2023; Strong et al., 2024). By dynamically assigning tasks to the most suitable agent, L2D ensures both accuracy and reliability, making it ideal for safety-critical domains.

Robustness in handling high-stakes decisions is, therefore, essential for such systems. However, existing L2D frameworks are typically designed under the assumption of clean, non-adversarial input data, leaving them highly susceptible to adversarial perturbations—subtle input manipulations that disrupt task allocation and alter decision boundaries. Unlike traditional machine learning systems, where adversarial attacks primarily affect prediction outputs (Goodfellow et al., 2015; Szegedy et al., 2014; Awasthi et al., 2023), L2D systems are susceptible to more sophisticated adversarial threats, including query redirection to less reliable agents or intentional agent overloading. These vulnerabilities severely impact performance, drive up operational costs, and compromise trust.

This paper addresses the important yet unexplored challenge of adversarial robustness in L2D systems. Inspired by adversarial attacks on classification (Goodfellow et al., 2015; Madry et al., 2018; Gowal et al., 2020), we introduce two novel attack strategies tailored to *two-stage* L2D: *untargeted attacks*, which disrupt agent allocation, and *targeted attacks*, which redirect queries to specific agents. To counter these attacks, we propose a robust family of surrogate losses based on cross-entropy (Mao et al., 2023a; 2024d; Montreuil et al., 2025c), designed to ensure robustness in classification, regression, and multi-task settings. Building upon advances in consistency theory for adversarial robustness (Bao et al., 2020; Awasthi et al., 2022; 2023; Mao et al., 2023b), we establish both Bayes-consistency and $(\mathcal{R}, \mathcal{G})$ consistency for our surrogate losses, enabling reliable task allocation even under adversarial scenarios. Our algorithm, **SARD**, leverages these guarantees while preserving convexity.

[1]School of Computing, National University of Singapore, Singapore [2]Institute for Infocomm Research, Agency for Science, Technology and Research, Singapore [3]CNRS@CREATE LTD, 1 Create Way, Singapore [4]IRIT, Université de Toulouse, CNRS, Toulouse INP, Toulouse, France [5]IPAL, IRL2955, Singapore. Correspondence to: Yannis Montreuil <yannis.montreuil@u.nus.edu>.

*Proceedings of the 42$^{nd}$ International Conference on Machine Learning*, Vancouver, Canada. PMLR 267, 2025. Copyright 2025 by the author(s).

Our key contributions are:

1. We introduce two new adversarial attack strategies tailored to *two-stage* L2D: *targeted* attacks that manipulate routing toward specific agents, and *untargeted* attacks that disrupt optimal allocation.

2. We propose a robust family of cross-entropy-based surrogate losses that are Bayes-consistent and $(\mathcal{R}, \mathcal{G})$-consistent, enabling reliable allocation in classification, regression, and multi-task settings. These losses are the foundation of our convex and efficient algorithm, SARD.

3. We empirically demonstrate that our attacks reveal critical vulnerabilities in state-of-the-art *two-stage* L2D systems, while our proposed method SARD consistently withstands these attacks and achieves robust performance across diverse tasks.

This work lays the theoretical and algorithmic foundations for adversarially robust Learning-to-Defer systems.

## 2. Related Works

**Learning-to-Defer.** Learning-to-Defer addresses the problem of allocating decisions between an automated model and offline decision-makers. In contrast to classical selective prediction—which simply rejects uncertain inputs without further action (Chow, 2003; Bartlett & Wegkamp, 2008; Cortes et al., 2016; Geifman & El-Yaniv, 2017)—L2D explicitly learns to defer to experts based on their distribution. Existing approaches fall into two broad categories: *one-stage* methods, where the predictor and deferral mechanism are trained jointly, and *two-stage* methods, where the deferral policy is optimized post hoc, assuming access to frozen predictions from pre-trained agents. This paper focuses on the two-stage setting.

The *one-stage* formulation was initiated by Madras et al. (2018), who jointly trained a predictor and a rejector in the binary setting. A major theoretical milestone was achieved by Mozannar & Sontag (2020), who established the first Bayes-consistent L2D method for multiclass classification, ensuring asymptotically optimal delegation. This was followed by Verma et al. (2023), who introduced a one-versus-all surrogate formulation that also achieves Bayes-consistency and was later generalized to broader surrogate families by Charusaie et al. (2022). To improve calibration across agents, Cao et al. (2023) proposed an asymmetric softmax surrogate that addresses the shortcomings identified in both Mozannar & Sontag (2020) and Verma et al. (2023). However, Mozannar et al. (2023) showed that these methods are not realizable-$\mathcal{H}$-consistent, potentially degrading performance when the hypothesis class is restricted. This issue

was tackled by Mao et al. (2025a; 2024c), who extended the theory to cover the realizable setting under specific conditions.

In the *two-stage* regime, where experts and the main model are fixed and trained offline, Narasimhan et al. (2022) introduced a post-hoc deferral strategy. Building on this, Mao et al. (2023a) provided the first Bayes- and $\mathcal{H}$-consistent formulation for classification, which was later extended to regression by Mao et al. (2024d). Finally, Montreuil et al. (2025c) generalized the two-stage framework to multi-task learning scenarios and Montreuil et al. (2025a) to top-$k$ experts.

**Adversarial Robustness:** The robustness of neural networks against adversarial perturbations has been extensively studied, with foundational work highlighting their vulnerabilities (Biggio et al., 2013; Szegedy et al., 2014; Goodfellow et al., 2015; Madry et al., 2018). A key focus in recent research has been on developing consistency frameworks for formulating robust defenses. Bao et al. (2020) proposed a Bayes-consistent surrogate loss tailored for adversarial training, which was further analyzed and extended in subsequent works (Meunier et al., 2022; Awasthi et al., 2021). Beyond Bayes-consistency, $\mathcal{H}$-consistency has been explored to address robustness in diverse settings. Notably, Awasthi et al. (2022) derived $\mathcal{H}$-consistency bounds for several surrogate families, and Mao et al. (2023b) conducted an in-depth analysis of the cross-entropy family. Building on these theoretical advancements, Awasthi et al. (2023) introduced a smooth algorithm that leverages consistency guarantees to enhance robustness in adversarial settings.

Our work builds upon recent advancements in consistency theory to further improve adversarial robustness in two-stage L2D.

## 3. Preliminaries

**Multi-task scenario.** We consider a multi-task setting that addresses both classification and regression problems simultaneously. Let $\mathcal{X}$ denote the input space, $\mathcal{Y} = \{1, \ldots, n\}$ represent the set of $n$ distinct classes for classification, and $\mathcal{T} \subseteq \mathbb{R}$ denote the target space for regression. Each data point is represented by a triplet $z = (x, y, t) \in \mathcal{Z}$, where $\mathcal{Z} = \mathcal{X} \times \mathcal{Y} \times \mathcal{T}$. We assume the data is drawn independently and identically distributed (i.i.d.) from an underlying distribution $\mathcal{D}$ over $\mathcal{Z}$. To model this multi-task problem, we introduce a *backbone* $w \in \mathcal{W}$, which acts as a shared feature extractor. The backbone maps inputs $x \in \mathcal{X}$ to a latent feature representation $q \in \mathcal{Q}$, via the function $w : \mathcal{X} \to \mathcal{Q}$. Building upon this backbone, we define a *classifier* $h \in \mathcal{H}$, representing all possible classification heads. Formally, $h : \mathcal{Q} \times \mathcal{Y} \to \mathbb{R}$ is a scoring function, with predictions computed as $h(q) = \arg\max_{y \in \mathcal{Y}} h(q, y)$.

Similarly, we define a *regressor* $f \in \mathcal{F}$, which maps latent features to real-valued targets, $f : \mathcal{Q} \to \mathcal{T}$. These components are integrated into a single multi-head network $g \in \mathcal{G}$, defined as $\mathcal{G} = \{g : g(x) = (h \circ w(x), f \circ w(x)) \mid w \in \mathcal{W}, h \in \mathcal{H}, f \in \mathcal{F}\}$.

**Consistency in classification.** In classification, the primary objective is to identify a classifier $h \in \mathcal{H}$ in the specific case where $w(x)$ is the identity function, such that $h(x) = \arg\max_{y \in \mathcal{Y}} h(x, y)$. This classifier should minimize the true error $\mathcal{E}_{\ell_{01}}(h)$, defined as $\mathcal{E}_{\ell_{01}}(h) = \mathbb{E}_{(x,y)}[\ell_{01}(h, x, y)]$. The Bayes-optimal error is expressed as $\mathcal{E}^{B}_{\ell_{01}}(\mathcal{H}) = \inf_{h \in \mathcal{H}} \mathcal{E}_{\ell_{01}}(h)$. However, minimizing $\mathcal{E}_{\ell_{01}}(h)$ directly is challenging due to the non-differentiability of the *true multiclass* 0-1 loss (Zhang, 2002; Steinwart, 2007; Awasthi et al., 2022; Mao et al., 2025c; Zhong, 2025; Mao et al., 2024e; Cortes et al., 2025). To address this challenge, surrogate losses are employed as convex, non-negative upper bounds on $\ell_{01}$. A notable family of multiclass surrogate losses is the comp-sum (Mohri et al., 2012; Mao et al., 2023b), which we refer to as a family of *multiclass surrogate* losses:

$$\Phi^{u}_{01}(h, x, y) = \Psi^{u}\Big( \sum_{y' \neq y} \Psi_{\mathrm{e}}(h(x, y) - h(x, y')) \Big), \quad (1)$$

where $\Psi_{\mathrm{e}}(v) = \exp(-v)$, which defines the cross-entropy family. For $u > 0$, the transformation is given by:

$$\Psi^{u}(v) = \begin{cases} \log(1 + v) & \text{if } u = 1, \\ \frac{1}{1-u}\left[(1-v)^{1-u} - 1\right] & \text{if } u > 0 \wedge u \neq 1. \end{cases} \quad (2)$$

This formulation generalizes several well-known loss functions, including the sum-exponential loss (Weston & Watkins, 1998), logistic loss (Ohn Aldrich, 1997), generalized cross-entropy (Zhang & Sabuncu, 2018), and mean absolute error loss (Ghosh et al., 2017). The corresponding true error for $\Phi^{u}_{01}$ is defined as $\mathcal{E}_{\Phi^{u}_{01}}(h) = \mathbb{E}_{(x,y)}[\Phi^{u}_{01}(h, x, y)]$, with its optimal value expressed as $\mathcal{E}^{*}_{\Phi^{u}_{01}}(\mathcal{H}) = \inf_{h \in \mathcal{H}} \mathcal{E}_{\Phi^{u}_{01}}(h)$.

A key property of a surrogate loss is *Bayes-consistency*, which ensures that minimizing the surrogate excess risk leads to minimizing the true excess risk (Zhang, 2002; Bartlett et al., 2006; Steinwart, 2007; Tewari & Bartlett, 2005; Cortes et al., 2024; Mao et al., 2025b). Formally, $\Phi^{u}_{01}$ is Bayes-consistent with respect to $\ell_{01}$ if, for any sequence $\{h_k\}_{k \in \mathbb{N}} \subset \mathcal{H}$, the following implication holds:

$$\begin{aligned} & \mathcal{E}_{\Phi^{u}_{01}}(h_k) - \mathcal{E}^{*}_{\Phi^{u}_{01}}(\mathcal{H}) \xrightarrow{k \to \infty} 0 \\ & \implies \mathcal{E}_{\ell_{01}}(h_k) - \mathcal{E}^{B}_{\ell_{01}}(\mathcal{H}) \xrightarrow{k \to \infty} 0. \end{aligned} \quad (3)$$

This property typically assumes $\mathcal{H} = \mathcal{H}_{\mathrm{all}}$, which may not hold for restricted hypothesis classes such as $\mathcal{H}_{\mathrm{lin}}$ or $\mathcal{H}_{\mathrm{ReLU}}$ (Long & Servedio, 2013; Awasthi et al., 2022; Mao

et al., 2023c). To characterize consistency with a particular hypothesis set, Awasthi et al. (2022) introduced $\mathcal{H}$-consistency bounds, which rely on a non-decreasing function $\Gamma : \mathbb{R}^{+} \to \mathbb{R}^{+}$ and take the following form:

$$\begin{aligned} & \mathcal{E}_{\Phi^{u}_{01}}(h) - \mathcal{E}^{*}_{\Phi^{u}_{01}}(\mathcal{H}) + \mathcal{U}_{\Phi^{u}_{01}}(\mathcal{H}) \geq \\ & \Gamma\Big( \mathcal{E}_{\ell_{01}}(h) - \mathcal{E}^{B}_{\ell_{01}}(\mathcal{H}) + \mathcal{U}_{\ell_{01}}(\mathcal{H}) \Big), \end{aligned} \quad (4)$$

where the minimizability gap $\mathcal{U}_{\ell_{01}}(\mathcal{H})$ quantifies the difference between the best-in-class excess risk and the expected pointwise minimum error: $\mathcal{U}_{\ell_{01}}(\mathcal{H}) = \mathcal{E}^{B}_{\ell_{01}}(\mathcal{H}) - \mathbb{E}_x\left[\inf_{h \in \mathcal{H}} \mathbb{E}_{y|x}\left[\ell_{01}(h, x, y)\right]\right]$. The gap vanishes when $\mathcal{H} = \mathcal{H}_{\mathrm{all}}$ (Steinwart, 2007; Awasthi et al., 2022; Mao et al., 2024b; 2023c; 2024a). In the asymptotic limit, inequality (4) ensures recovery of Bayes-consistency (3).

**Adversarial robustness.** Adversarially robust classification aims to train classifiers that are robust to small, imperceptible perturbations of the input (Goodfellow et al., 2015; Madry et al., 2018). The objective is to minimize the *true multiclass loss* $\ell_{01}$ evaluated on an adversarial input $x' = x + \delta$ (Gowal et al., 2020; Awasthi et al., 2022). A perturbation $\delta$ is constrained by its magnitude, and we define the adversarial region around $x$ as $B_p(x, \gamma) = \{x' \mid \|x' - x\|_p \leq \gamma\}$, where $\|\|_p$ is the $p$-norm and $\gamma \in (0, 1)$ specifies the maximum allowed perturbation.

**Definition 3.1** (Adversarial True Multiclass Loss)**.** The *adversarial true multiclass loss* $\widetilde{\ell}_{01} : \mathcal{H} \times \mathcal{X} \times \mathcal{Y} \to \{0, 1\}$ is given by:

$$\widetilde{\ell}_{01}(h, x, y) = \sup_{x' \in B_p(x, \gamma)} \ell_{01}(h(x'), y).$$

Similarly to classification, minimizing $\widetilde{\ell}_{01}$ is computationally infeasible (Zhang & Agarwal, 2020; Bartlett et al., 2006; Awasthi et al., 2022). To address this, we introduce the following surrogates.

**Definition 3.2** (Adversarial Margin Surrogates)**.** The family of *adversarial margin surrogate* losses $\widetilde{\Phi}^{\rho,u}_{01}$ from the comp-sum $\rho$-margin family, which approximate the *adversarial true multiclass loss* $\widetilde{\ell}_{01}$ is defined as:

$$\widetilde{\Phi}^{\rho,u}_{01}(h, x, y) = \sup_{x' \in B_p(x, \gamma)} \Psi^{u}\Big( \sum_{y' \neq y} \Psi_{\rho}(h(x', y') - h(x', y)) \Big).$$

Here, $\Psi^{u}$ and $\Psi_{\rho}$ represent transformations that characterize the behavior of the family. The piecewise-linear and non-convex transformation is defined as $\Psi_{\rho}(v) = \min\left\{\max\left(0, 1 - \frac{v}{\rho}\right), 1\right\}$. Recent studies have shown that algorithms based on smooth, regularized variants of the comp-sum $\rho$-margin losses achieve $\mathcal{H}$-consistency, thereby offering strong theoretical guarantees (Awasthi et al., 2022; 2023; Mao et al., 2023b).

**Two-Stage Learning-to-Defer.** The Learning-to-Defer (L2D) framework assigns each input $x \in \mathcal{X}$ to the most cost-effective *agent*, leveraging the strengths of multiple agents to improve task performance. Agents consist of a fixed primary model and a set of $J$ offline experts. We denote the full agent set as $\mathcal{A} = \{0\} \cup [J]$, where $j = 0$ refers to the primary model $g$ defined in Section 3. Each expert $M_j$ outputs a prediction pair $m_j(x) = (m_j^h(x), m_j^f(x))$, with $m_j^h(x) \in \mathcal{Y}$ a categorical prediction and $m_j^f(x) \in \mathcal{T}$ a regression output. The complete set of expert predictions is denoted $m(x) = (m_1(x), \ldots, m_J(x)) \in \mathcal{M}$. In the two-stage setting, all agents are trained offline, and only the allocation function is learned, keeping agent parameters fixed.

A rejector $r \in \mathcal{R}$, defined as $r : \mathcal{X} \times \mathcal{A} \to \mathbb{R}$, is trained to select the most suitable agent by computing $r(x) = \arg\max_{j \in \mathcal{A}} r(x, j)$, as proposed by Mao et al. (2024d; 2023a); Montreuil et al. (2025c).

**Definition 3.3** (Two-Stage L2D Losses). Let $x \in \mathcal{X}$ and $r \in \mathcal{R}$. The *true deferral loss* is defined as:

$$\ell_{\text{def}}(r, g, m, z) = \sum_{j=0}^{J} c_j(g(x), m_j(x), z) 1_{r(x)=j}.$$

Its convex, non-negative, upper-bound surrogate family is given by:

$$\Phi_{\text{def}}^u(r, g, m, z) = \sum_{j=0}^{J} \tau_j(g(x), m(x), z) \Phi_{01}^u(r, x, j),$$

where $c_j$ denotes the cost of assigning query $x$ to agent $j \in \mathcal{A}$ (Madras et al., 2018). If $r(x) = 0$, the decision is handled by the primary model $g$, which returns prediction $g(x) = (h(w(x)), f(w(x)))$ and incurs cost $c_0(g(x), z) = \psi(g(x), z)$, where $\psi(\cdot, \cdot) \in \mathbb{R}^+$ is a general loss function.

If $r(x) = j$ for some $j > 0$, the query is deferred to expert $j$, incurring a cost $c_j(m_j(x), z) = \psi(m_j(x), z) + \beta_j$, where $\beta_j \geq 0$ is the consultation cost. The complementary cost function used in the surrogate is:

$$\tau_j(g(x), m(x), z) = \sum_{i=0}^{J} c_i(g(x), m_i(x), z) 1_{i \neq j},$$

which matches the formulation from Mao et al. (2024d); Montreuil et al. (2025c). In the case of classification, $\psi$ corresponds to the standard 0–1 loss.

We illustrate how Learning-to-Defer works at inference in Figure 1.

## 4. Novel Adversarial Attacks on L2D

**Motivation.** The two-stage L2D framework is designed to route queries to the most accurate agents, ensuring opti-

mal decision-making (Narasimhan et al., 2022; Mao et al., 2023a; 2024d; Montreuil et al., 2025c). Despite its effectiveness, we show that this framework is inherently vulnerable to adversarial attacks that exploit its reliance on the rejector function—a key component responsible for query allocation. Given this critical role, our analysis concentrates on adversarial attacks and corresponding defenses targeting the rejector $r \in \mathcal{R}$, rather than individual agents. This emphasis is warranted by the fact that adversarial defenses for specific agents can typically be deployed offline within the two-stage L2D setup, whereas the rejector operates as a router at inference time, serving as the first layer in the architecture to direct queries to the appropriate agent. Moreover, evaluating the robustness of individual agents under adversarial conditions reduces to selecting the most robust agent, whereas ensuring robustness at the system level constitutes a fundamentally different challenge.

**Untargeted Attack.** In classification, the goal of an untargeted attack is to find a perturbed input $x' \in B_p(x, \gamma)$ that causes the classifier $h \in \mathcal{H}$ to misclassify the input (Goodfellow et al., 2015; Akhtar & Mian, 2018). Specifically, for a clean input $x \in \mathcal{X}$ such that the classifier correctly predicts $h(x) = y$, the attacker seeks a perturbed input $x'$ such that $h(x') \neq y$.

In the context of Learning-to-Defer, the attack extends beyond misclassification to compromising the decision allocation mechanism. Given an optimal agent $j^* \in \mathcal{A}$, the attacker aims to find an adversarial input $x' \in B_p(x, \gamma)$ such that $r(x') \neq j^*$, thereby resulting in a deferral to a suboptimal agent $j \in \mathcal{A} \setminus \{j^*\}$, leading to an increased loss. This gives rise to the following untargeted attack formulation:

**Definition 4.1** (Untargeted Attack in L2D). Let $x' \in B_p(x, \gamma)$ be an adversarial input, where $B_p(x, \gamma)$ denotes the $p$-norm ball of radius $\gamma$ centered at $x$. The untargeted attack that maximizes misallocation in L2D is defined as:

$$x' = \arg\max_{x' \in B_p(x, \gamma)} \sum_{j=0}^{J} \tau_j(g(x), m(x), z) \Phi_{01}^u(r, x', j)$$

Definition 4.1 characterizes an attack in which the adversary maximizes the overall surrogate loss incurred by misallocation, thereby causing the system to defer to an unintended agent. The adversarial input $x'$ is designed to maximize these surrogate losses. As a result, the attack increases the system's overall loss, leading to a significant degradation in performance. An illustration of this attack is provided in Appendix Figure 2.

**Targeted Attack.** Targeted attacks are often more impactful than untargeted ones, as they exploit specific system vulnerabilities to achieve precise adversarial goals (Akhtar

& Mian, 2018; Chakraborty et al., 2021). For example, in autonomous driving, a targeted attack could deliberately misclassify a stop sign as a speed limit sign, potentially leading to hazardous consequences. Such attacks exploit asymmetries in the task by forcing the classifier $h \in \mathcal{H}$ to predict a specific target class $y_t \in \mathcal{Y}$, thereby amplifying the risk of incorrect or dangerous decisions.

In the context of L2D, the attacker seeks to manipulate the system to assign a query $x' \in B_p(x, \gamma)$ to a predetermined expert $j_t \in \mathcal{A}$, rather than the optimal expert $j^* \in \mathcal{A}$. This targeted attack can be formally expressed as follows:

**Definition 4.2** (Targeted Attack in L2D). Let $x' \in B_p(x, \gamma)$, where $B_p(x, \gamma)$ denotes the $p$-norm ball of radius $\gamma$ centered at $x$. The targeted attack that biases the allocation of a query $x$ toward a predetermined expert $j_t \in \mathcal{A}$ is defined as

$$x' = \underset{x' \in B_p(x,\gamma)}{\arg\min} \ \tau_{j_t}(g(x), m(x), z)\Phi_{01}^u(r, x', j_t).$$

The adversarial input $x'$ minimizes the surrogate loss associated with the target agent $j_t$, thereby biasing the allocation process toward agent $j_t$ (Definition 4.2).

For instance, an attacker may have an affiliated partner $j_t$ among the system's agents. Suppose the system operates under a pay-per-query model—such as a specialist doctor in a medical decision-making system or a third-party service provider in an AI platform. By manipulating the allocation mechanism to systematically route more queries to $j_t$, the attacker artificially inflates the agent's workload, resulting in unjustified financial gains shared between the attacker and the affiliated expert through direct payments, revenue-sharing, or other collusive arrangements. An illustration of this attack is provided in Appendix Figure 3.

## 5. Adversarially Consistent Formulation for Two-Stage Learning-to-Defer

In this section, we introduce an adversarially consistent formulation of the two-stage L2D framework that ensures robustness against attacks while preserving optimal query allocation.

### 5.1. Novel Two-stage Learning-to-Defer Formulation

**Adversarial True Deferral Loss.** To defend against the novel attacks introduced in Section 4, we define the worst-case *adversarial true deferral loss*, $\widetilde{\ell}_{def} : \mathcal{R} \times \mathcal{G} \times \mathcal{M} \times \mathcal{Z} \to \mathbb{R}^+$, which quantifies the maximum incurred loss under adversarial perturbations. Specifically, for each $j \in \mathcal{A}$, an adversarial perturbation $\delta_j$ is applied, resulting in a perturbed input $x'_j = x + \delta_j$, where $x'_j \in B_p(x, \gamma)$ lies within an $\ell_p$-norm ball of radius $\gamma$.

**Definition 5.1** ($j$-th Adversarial True Multiclass Loss). We

define the $j$-th adversarial true multiclass loss as

$$\widetilde{\ell}_{01}^j(r, x, j) = \sup_{x'_j \in B_p(x,\gamma)} \ell_{01}(r(x'_j), j),$$

which captures the worst-case misclassification loss when deferring to agent $j$ under adversarial conditions. The adversarial true deferral loss is defined in Lemma 5.2.

**Lemma 5.2** (Adversarial True Deferral Loss). *Let $x \in \mathcal{X}$ denote the clean input, $c_j$ the cost associated with agent $j \in \mathcal{A}$, and $\tau_j$ the aggregated cost. The adversarial true deferral loss $\widetilde{\ell}_{def}$ is defined as:*

$$\widetilde{\ell}_{def}(r, g, m, z) = \sum_{j=0}^{J} \tau_j(g(x), m(x), z)\widetilde{\ell}_{01}^j(r, x, j)$$

$$+ (1 - J)\sum_{j=0}^{J} c_j(g(x), m_j(x), z).$$

See Appendix D.1 for the proof of Lemma 5.2. The attacker's objective is to compromise the allocation process by identifying perturbations $\delta_j$ that maximize the loss for each agent $j \in \mathcal{A}$. Notably, the costs $c_j$ and $\tau_j$ are evaluated on the clean input $x$, as the agents' predictions remain unaffected by the perturbations (see Section 4).

*Remark* 5.3. Simply applying adversarial evaluation to the entire two-stage loss (Definition 3.3) does not account for the worst-case scenario captured in Lemma 5.2. This point is further discussed in Appendix D.1.

As with standard classification losses, minimizing the adversarial true deferral loss in Lemma 5.2 is NP-hard (Zhang, 2002; Bartlett et al., 2006; Steinwart, 2007; Awasthi et al., 2022). Therefore, as in classification problems, we approximate this discontinuous loss using surrogates.

**Adversarial Margin Deferral Surrogate Losses.** In the formulation of the *adversarial true deferral loss* (Lemma 5.2), discontinuities arise from the use of the indicator function in its definition. To approximate this discontinuity, we build on recent advancements in consistency theory for adversarially robust classification (Bao et al., 2020; Awasthi et al., 2022; 2023; Mao et al., 2023b) and propose a continuous, upper-bound surrogate family to approximate it.

**Definition 5.4** ($j$-th Adversarial Margin Surrogates). We define the $j$-th adversarial margin surrogate family as

$$\widetilde{\Phi}_{01}^{\rho,u,j}(r, x, j) = \sup_{x'_j \in B_p(x,\gamma)} \Psi^u\left(\sum_{j' \neq j} \Psi_\rho\big(r(x'_j, j') - r(x'_j, j)\big)\right)$$

where $\Psi^u$ and $\Psi_\rho$ are defined in Equation (3.2). Based on this definition, we derive the *adversarial margin deferral surrogate* losses as:

**Lemma 5.5** (Adversarial Margin Deferral Surrogate Losses). *Let $x \in \mathcal{X}$ denote the clean input and $\tau_j$ the aggregated cost. The adversarial margin deferral surrogate losses $\widetilde{\Phi}_{def}^{\rho,u}$ are then defined as:*

$$\widetilde{\Phi}_{def}^{\rho,u}(r, g, m, z) = \sum_{j=0}^{J} \tau_j(g(x), m(x), z)\widetilde{\Phi}_{01}^{\rho,u,j}(r, x, j).$$

The proof is provided in Appendix D.2. A key limitation of the *adversarial margin deferral surrogate* family lies in the non-convexity of the *j-th adversarial margin surrogate loss* $\widetilde{\Phi}_{01}^{\rho,u,j}$, induced by the piecewise nature of $\Psi_\rho(v)$, which complicates efficient optimization.

**Adversarial Smooth Deferral Surrogate Losses.** Prior work by Awasthi et al. (2023); Mao et al. (2023b) shows that the non-convex *adversarial margin surrogate* family can be approximated by a smooth and convex surrogate. Building on their results, we introduce the *smooth adversarial surrogate* family, denoted as $\widetilde{\Phi}_{01}^{smth,u} : \mathcal{R} \times \mathcal{X} \times \mathcal{A} \to \mathbb{R}^+$, which approximates the supremum term in Lemma 5.2. Importantly, $\widetilde{\Phi}_{01}^{smth,u}$ serves as a convex, non-negative upper bound for the *j-th adversarial margin surrogate* family, satisfying $\widetilde{\Phi}_{01}^{\rho,u,j} \leq \widetilde{\Phi}_{01}^{smth,u}$. We derive this surrogate in Lemma 5.6.

**Lemma 5.6** (Smooth Adversarial Surrogate Losses). *Let $x \in \mathcal{X}$ denote the clean input and hyperparameters $\rho > 0$ and $\nu > 0$. The smooth adversarial surrogate losses are defined as:*

$$\widetilde{\Phi}_{01}^{smth,u}(r, x, j) = \Phi_{01}^u\left(\frac{r}{\rho}, x, j\right)$$
$$+ \nu \sup_{x_j' \in B_p(x,\gamma)} \|\overline{\Delta}_r(x_j', j) - \overline{\Delta}_r(x, j)\|_2.$$

The proof is provided in Appendix D.3. For $x \in \mathcal{X}$, define $\Delta_r(x, j, j') = r(x, j) - r(x, j')$, and let $\overline{\Delta}_r(x, j) \in \mathbb{R}^J$ denote the vector of pairwise differences $\big(\Delta_r(x, j, 0), \ldots, \Delta_r(x, j, j - 1), \Delta_r(x, j, j + 1), \ldots, \Delta_r(x, j, J)\big)$. The first term $\Phi_{01}^u\left(\frac{r}{\rho}, x, j\right)$ corresponds to the *multiclass surrogate* losses modulated by the coefficient $\rho$. The second term introduces a smooth adversarial component over perturbations $x_j' \in B_p(x, \gamma)$, scaled by the coefficient $\nu$ (Awasthi et al., 2023; Mao et al., 2023b). The hyperparameters $(\rho, \nu)$ are typically selected via cross-validation to balance allocation performance and robustness.

Using this surrogate, we define the *smooth adversarial deferral surrogate* (SAD) family $\widetilde{\Phi}_{def}^{smth,u} : \mathcal{R} \times \mathcal{G} \times \mathcal{M} \times \mathcal{Z} \to \mathbb{R}^+$, which is convex, non-negative, and upper-bounds $\widetilde{\ell}_{def}$ by construction. Its formal definition is as follows:

**Lemma 5.7** (SAD: **S**mooth **A**dversarial **D**eferral Surrogate Losses). *Let $x \in \mathcal{X}$ denote the clean input and $\tau_j$ the*

*aggregated cost. Then, the smooth adversarial surrogate family (SAD) $\widetilde{\Phi}_{def}^{smth,u}$ is defined as:*

$$\widetilde{\Phi}_{def}^{smth,u}(r, g, m, z) = \sum_{j=0}^{J} \tau_j(g(x), m(x), z)\widetilde{\Phi}_{01}^{smth,u}(r, x, j)$$

The proof is provided in Appendix D.4. While the SAD family offers a smooth and computationally efficient approximation of the *adversarial true deferral loss*, the question of its consistency remains a central concern.

### 5.2. Theoretical Guarantees

To establish the theoretical foundations of SAD, we first prove that the family of *adversarial margin deferral surrogates* $\widetilde{\Phi}_{def}^{\rho,u}$ is Bayes-consistent and $(\mathcal{R}, \mathcal{G})$-consistent (Lemma 5.5). Wang & Scott (2023) introduced sufficient conditions for a surrogate loss to satisfy Bayes-consistency. These conditions—Gamma-PD (Definition 3.1) and Phi-NDZ (Definition 3.2)—are jointly sufficient to apply their general Theorem 2.6.

While our surrogate satisfies the Gamma-PD property, it fails to meet the Phi-NDZ criterion because it is not differentiable on all of $\mathbb{R}$. As a consequence, Theorem 2.6 from Wang & Scott (2023) cannot be invoked directly to establish classification-calibration. This limitation motivates a dedicated analysis of the consistency properties of our surrogates. We show that both Bayes- and $(\mathcal{R}, \mathcal{G})$-consistency can nonetheless be established for the class $\widetilde{\Phi}_{def}^{\rho,u}$, and that these guarantees extend to the regularized empirical variant used in our algorithmic implementation, referred to as SARD.

$(\mathcal{R}, \mathcal{G})$**-consistency bounds of $\widetilde{\Phi}_{def}^{\rho,u}$.** A key step toward establishing theoretical guarantees is to demonstrate the $\mathcal{R}$-consistency of the *j-th adversarial margin surrogate* losses $\widetilde{\Phi}_{01}^{\rho,u,j}$, under the assumption that $\mathcal{R}$ is symmetric and that there exists a rejector $r \in \mathcal{R}$ that is *locally $\rho$-consistent*.

**Definition 5.8** (Locally $\rho$-consistent). *A hypothesis set $\mathcal{R}$ is locally $\rho$-consistent if, for any $x \in \mathcal{X}$, there exists a hypothesis $r \in \mathcal{R}$ such that:*

$$\inf_{x' \in B_p(x,\gamma)} |r(x', i) - r(x', j)| \geq \rho,$$

*where $\rho > 0$, $i \neq j \in \mathcal{A}$, and $x' \in B_p(x, \gamma)$. Moreover, the ordering of the values $\{r(x', j)\}$ is preserved with respect to $\{r(x, j)\}$ for all $x' \in B_p(x, \gamma)$.*

As shown by Awasthi et al. (2022); Mao et al. (2023b); Awasthi et al. (2023), common hypothesis classes—including linear models, neural networks, and the set of all measurable functions—are locally $\rho$-consistent for some $\rho > 0$. Consequently, the guarantees established

in Lemma 5.9 are broadly applicable in practice. See Appendix D.5 for the proof.

**Lemma 5.9** ($\mathcal{R}$-consistency bounds for $\widetilde{\Phi}_{01}^{\rho,u,j}$). *Assume $\mathcal{R}$ is symmetric and locally $\rho$-consistent. Then, for the agent set $\mathcal{A}$, any hypothesis $r \in \mathcal{R}$, and any distribution $\mathcal{P}$ with probabilities $p = (p_0, \ldots, p_J) \in \Delta^{|\mathcal{A}|}$, the following inequality holds:*

$$\sum_{j \in \mathcal{A}} p_j \widetilde{\ell}_{01}^j(r, x, j) - \inf_{r \in \mathcal{R}} \sum_{j \in \mathcal{A}} p_j \widetilde{\ell}_{01}^j(r, x, j) \leq$$

$$\Psi^u(1) \Big( \sum_{j \in \mathcal{A}} p_j \widetilde{\Phi}_{01}^{\rho,u,j}(r, x, j) - \inf_{r \in \mathcal{R}} \sum_{j \in \mathcal{A}} p_j \widetilde{\Phi}_{01}^{\rho,u,j}(r, x, j) \Big).$$

Lemma 5.9 establishes the consistency of the *$j$-th adversarial margin surrogate* family $\widetilde{\Phi}_{01}^{\rho,u,j}$ under adversarial perturbations defined for each $j \in \mathcal{A}$, weighted by distributional probabilities $p_j \in \Delta^{|\mathcal{A}|}$. This result sets our work apart from prior approaches (Mao et al., 2023b; Awasthi et al., 2023; 2022), which do not account for adversarial inputs at the per-agent distributional level. By addressing this limitation, Lemma 5.9 provides a crucial theoretical guarantee: it shows that the surrogate losses $\widetilde{\Phi}_{01}^{\rho,u,j}$ align with the true adversarial losses $\widetilde{\ell}_{01}^j$ under appropriate conditions.

Building on this result, we establish the Bayes- and $(\mathcal{R}, \mathcal{G})$-consistency of the *adversarial margin deferral surrogate* losses. The proof of Theorem 5.10 is provided in Appendix D.6.

**Theorem 5.10** ($(\mathcal{R}, \mathcal{G})$-consistency bounds of $\widetilde{\Phi}_{\text{def}}^{\rho,u}$). *Let $\mathcal{R}$ be symmetric and locally $\rho$-consistent. Then, for the agent set $\mathcal{A}$, any hypothesis $r \in \mathcal{R}$, and any distribution $\mathcal{D}$, the following holds for a multi-task model $g \in \mathcal{G}$:*

$$\mathcal{E}_{\widetilde{\ell}_{def}}(r, g) - \mathcal{E}_{\widetilde{\ell}_{def}}^B(\mathcal{R}, \mathcal{G}) + \mathcal{U}_{\widetilde{\ell}_{def}}(\mathcal{R}, \mathcal{G}) \leq$$

$$\Psi^u(1) \Big( \mathcal{E}_{\widetilde{\Phi}_{def}^{\rho,u}}(r) - \mathcal{E}_{\widetilde{\Phi}_{def}^{\rho,u}}^*(\mathcal{R}) + \mathcal{U}_{\widetilde{\Phi}_{def}^{\rho,u}}(\mathcal{R}) \Big)$$

$$+ \mathcal{E}_{c_0}(g) - \mathcal{E}_{c_0}^B(\mathcal{G}) + \mathcal{U}_{c_0}(\mathcal{G}).$$

Theorem 5.10 establishes that the adversarial margin deferral surrogate family $\widetilde{\Phi}_{\text{def}}^{\rho,u}$ is consistent with the adversarial true deferral loss $\widetilde{\ell}_{\text{def}}$. When the minimizability gaps vanish—i.e., when $\mathcal{R} = \mathcal{R}_{\text{all}}$ and $\mathcal{G} = \mathcal{G}_{\text{all}}$ (Steinwart, 2007; Awasthi et al., 2022)—the bound reduces to:

$$\mathcal{E}_{\widetilde{\ell}_{def}}(r, g) - \mathcal{E}_{\widetilde{\ell}_{def}}^B(\mathcal{R}, \mathcal{G}) \leq \mathcal{E}_{c_0}(g) - \mathcal{E}_{c_0}^B(\mathcal{G})$$

$$+ \Psi^u(1) \Big( \mathcal{E}_{\widetilde{\Phi}_{def}^{\rho,u}}(r) - \mathcal{E}_{\widetilde{\Phi}_{def}^{\rho,u}}^*(\mathcal{R}) \Big).$$

Assume that after offline training, the multi-task model satisfies $\mathcal{E}_{c_0}(g) - \mathcal{E}_{c_0}^B(\mathcal{G}) \leq \epsilon_0$, and the trained rejector satisfies $\mathcal{E}_{\widetilde{\Phi}_{def}^{\rho,u}}(r) - \mathcal{E}_{\widetilde{\Phi}_{def}^{\rho,u}}^*(\mathcal{R}) \leq \epsilon_1$. Then, the excess adversarial deferral risk is bounded as:

$$\mathcal{E}_{\widetilde{\ell}_{def}}(r, g) - \mathcal{E}_{\widetilde{\ell}_{def}}^B(\mathcal{R}, \mathcal{G}) \leq \epsilon_0 + \Psi^u(1)\epsilon_1,$$

thereby establishing both Bayes-consistency and $(\mathcal{R}, \mathcal{G})$-consistency of the surrogate losses $\widetilde{\Phi}_{\text{def}}^{\rho,u}$.

We now introduce the *smooth adversarial regularized deferral* (SARD) algorithm, which extends the standard SAD framework by incorporating a regularization term. SARD retains the theoretical guarantees of $\widetilde{\Phi}_{\text{def}}^{\rho,u}$, including consistency and minimizability under the same assumptions (Mao et al., 2023b; Awasthi et al., 2023).

**Guarantees for SARD.** We now extend our theoretical guarantees to the *smooth adversarial deferral surrogate* (SAD) family $\widetilde{\Phi}_{\text{def}}^{\text{smth},u}$, leveraging the inequality $\widetilde{\Phi}_{01}^{\text{smth},u} \geq \widetilde{\Phi}_{01}^{\rho,u,j}$. Under the same assumptions as in Theorem 5.10, we obtain the following bound:

**Corollary 5.11** (Guarantees for SAD). *Assume $\mathcal{R}$ is symmetric and locally $\rho$-consistent. Then, for the agent set $\mathcal{A}$, any hypothesis $r \in \mathcal{R}$, and any distribution $\mathcal{D}$, the following holds for any multi-task model $g \in \mathcal{G}$:*

$$\mathcal{E}_{\widetilde{\ell}_{def}}(r, g) - \mathcal{E}_{\widetilde{\ell}_{def}}^B(\mathcal{R}, \mathcal{G}) + \mathcal{U}_{\widetilde{\ell}_{def}}(\mathcal{R}, \mathcal{G})$$

$$\leq \Psi^u(1) \Big( \mathcal{E}_{\widetilde{\Phi}_{def}^{smth,u}}(r) - \mathcal{E}_{\widetilde{\Phi}_{def}^{\rho,u}}^*(\mathcal{R}) + \mathcal{U}_{\widetilde{\Phi}_{def}^{\rho,u}}(\mathcal{R}) \Big)$$

$$+ \mathcal{E}_{c_0}(g) - \mathcal{E}_{c_0}^B(\mathcal{G}) + \mathcal{U}_{c_0}(\mathcal{G}).$$

Corollary 5.11 shows that SAD satisfies similar consistency guarantees under the same assumptions as its non-smooth counterpart. In particular, the minimizability gap vanishes when $\mathcal{R} = \mathcal{R}_{\text{all}}$ and $\mathcal{G} = \mathcal{G}_{\text{all}}$, further justifying the use of SAD in practice.

Motivated by this theoretical foundation, we introduce a learning algorithm that minimizes a regularized empirical approximation of SAD, recovering guarantees established in both Theorem 5.10 and Corollary 5.11. We refer to this algorithm as SARD (**S**mooth **A**dversarial **R**egularized **D**eferral).

**Proposition 5.12** (SARD: **S**mooth **A**dversarial **R**egularized **D**eferral Algorithm). *Assume $\mathcal{R}$ is symmetric and locally $\rho$-consistent. Let $\Omega : \mathcal{R} \to \mathbb{R}^+$ be a regularizer and $\eta > 0$ a hyperparameter. The regularized empirical risk minimization objective solved by SARD is:*

$$\min_{r \in \mathcal{R}} \left[ \frac{1}{K} \sum_{k=1}^K \widetilde{\Phi}_{def}^{smth,u}(r, g, m, z_k) + \eta \, \Omega(r) \right].$$

The pseudo-code for SARD is provided in Appendix C.

### 5.3. Computational complexity of SARD

Having established statistical guarantees, we now quantify the complexity cost of SARD. Because training time can become a bottleneck for large-scale models, especially when

inner maximizations are involved, we spell out the precise forward/backward counts so that future work can reproduce or improve our implementation.

**Proposition 5.13** (Epoch cost of SARD). *Process $n$ training examples in mini-batches of size $B$. Let $\mathcal{A} = \{0\} \cup [J]$ be the set of deferral outcomes ($|\mathcal{A}| = J + 1$) and let $T_a \in \mathbb{N}$ be the number of projected-gradient steps used in the inner maximizations (PGD$T_a$). Write $C_{\mathrm{fwd}}$ and $C_{\mathrm{bwd}}$ for the costs of one forward and one backward pass through the rejector network $r_\theta$. Then one epoch of SARD minimization incurs*

$$n \left(1 + |\mathcal{A}| \, T_a\right)\left(C_{\mathrm{fwd}} + C_{\mathrm{bwd}}\right)$$

*network traversals, while the peak memory equals that of a single forward–backward pass plus the storage of one adversarial copy of each input.*

*Proof.* Per mini-batch SARD executes (i) one clean forward of $r_\theta$; (ii) for each $j \in \mathcal{A}$ and each of the $T_a$ PGD steps, one forward and one backward of $r_\theta$; (iii) one backward pass to update $\theta$. This totals $2(1 + |\mathcal{A}| T_a)$ network traversals. Multiplying by the $\lceil n/B \rceil$ batches yields (5.13). Because PGD steps are processed sequentially, only the activations of the clean forward plus those of the current PGD iteration must reside in memory, giving the stated peak footprint. $\square$

**Comparison with Two-Stage L2D losses.** For the standard convex deferral loss $\Phi_{\mathrm{def}}^u = \sum_{j \in \mathcal{A}} \tau_j \, \Phi_{01}^u\left(r, x, j\right)$ the inner maximizations is absent ($T_a = 0$), so its epoch cost is simply $n\left(C_{\mathrm{fwd}} + C_{\mathrm{bwd}}\right)$. Equation (5.13) therefore shows that SARD introduces a purely multiplicative overhead factor $1 + |\mathcal{A}| \, T_a$.

Although the factor in (5.13) appears large, the absolute overhead is modest on contemporary hardware. Moreover, the overhead scales *linearly* with both the number of experts $|\mathcal{A}|$ and the attack depth $T_a$, providing clear knobs for future work to trade robustness for wall-time. These observations guide our experimental choices in the next section.

## 6. Experiments

We evaluate the robustness of SARD against state-of-the-art two-stage L2D frameworks across three tasks: classification, regression, and multi-task learning. Our experiments reveal that while existing baselines achieve slightly higher performance under clean conditions, they suffer from severe performance degradation under adversarial attacks. In contrast, SARD consistently maintains high performance, demonstrating superior robustness to both untargeted and targeted attacks. To the best of our knowledge, this is the first study to address adversarial robustness within the context of Learning-to-Defer.

### 6.1. Multiclass Classification Task

We compare our robust SARD formulation against the method introduced by Mao et al. (2023a) on the CIFAR-100 dataset (Krizhevsky, 2009).

**Setting:** Categories were assigned to three experts with a correctness probability $p = 0.94$, while the remaining probability was uniformly distributed across the other categories, following the approach in (Mozannar & Sontag, 2020; Verma et al., 2023; Cao et al., 2023). To further evaluate robustness, we introduced a weak expert $M_3$, with only a few assigned categories, and assumed that the attacker is aware of this weakness. Agent costs are defined as $c_0(h(x), y) = \ell_{01}(h(x), y)$ for the model and $c_{j>0}(m_j^h(x), y) = \ell_{01}(m_j^h(x), y)$, aligned with (Mozannar & Sontag, 2020; Mozannar et al., 2023; Verma et al., 2023; Cao et al., 2023; Mao et al., 2023a). Both the model and the rejector were implemented using ResNet-4 (He et al., 2015). The agents' performance, additional training details, and experimental results are provided in Appendix E.1.

| Baseline | Clean | Untarg. | Targ. $M_1$ | Targ. $M_2$ | Targ. $M_3$ |
|---|---|---|---|---|---|
| Mao et al. (2023a) | $72.8 \pm 0.4$ | $17.2 \pm 0.2$ | $54.4 \pm 0.1$ | $45.4 \pm 0.1$ | $13.4 \pm 0.1$ |
| Our | $67.0 \pm 0.4$ | $49.8 \pm 0.3$ | $62.4 \pm 0.3$ | $62.1 \pm 0.2$ | $64.8 \pm 0.3$ |

*Table 1.* Comparison of accuracy results between the proposed SARD and the baseline (Mao et al., 2023a) on the CIFAR-100 validation set, including clean and adversarial scenarios.

**Results:** The results in Table 1 underscore the robustness of our proposed SARD algorithm. While the baseline achieves a higher clean accuracy (72.8% vs. 67.0%), this comes at the cost of extreme vulnerability to adversarial attacks. In contrast, SARD prioritizes robustness, significantly outperforming the baseline under adversarial conditions. Specifically, in the presence of untargeted attacks, SARD retains an accuracy of 49.8%, a 2.9 times improvement over the baseline's sharp decline to 17.2%. Similarly, under targeted attacks aimed at the weak expert $M_3$, our method achieves 64.8% accuracy, a stark contrast to the baseline's 13.4%, highlighting SARD's ability to counteract adversarial exploitation of weak experts. These findings validate the efficacy of SARD in preserving performance across diverse attack strategies.

### 6.2. Regression Task

We evaluate the performance of SARD against the method proposed by Mao et al. (2024d) using the California Housing dataset involving median house price prediction (Kelley Pace & Barry, 1997).

**Setting:** We train three experts, each implemented as an MLP, specializing in a specific subset of the dataset based

on a predefined localization criterion. Among these, expert $M_3$ is designed to specialize in a smaller region, resulting in comparatively weaker overall performance. Agent costs for regression are defined as $c_0(f(x), t) = \text{RMSE}(f(x), t)$ for the model and $c_{j>0}(m_j^f(x), t) = \text{RMSE}(m_j^f(x), t)$, aligned with (Mao et al., 2024d). Both the model and the rejector are trained on the full dataset using MLPs. We provide detailed agent performance results, training procedures, and additional experimental details in Appendix E.2.

| Baseline | Clean | Untarg. | Targ. $M_1$ | Targ. $M_2$ | Targ. $M_3$ |
|---|---|---|---|---|---|
| Mao et al. (2024d) | $0.17 \pm 0.01$ | $0.29 \pm 0.3$ | $0.40 \pm 0.02$ | $0.21 \pm 0.01$ | $0.41 \pm 0.05$ |
| Our | $0.17 \pm 0.01$ | $0.17 \pm 0.01$ | $0.18 \pm 0.01$ | $0.18 \pm 0.01$ | $0.18 \pm 0.01$ |

*Table 2.* Performance comparison of SARD with the baseline (Mao et al., 2024d) on the California Housing dataset. The table reports Root Mean Square Error (RMSE).

**Results:** Table 2 presents the comparative performance of the baseline and SARD under clean and adversarial conditions. Under clean settings, both approaches achieve similar performance with an RMSE of 0.17. However, under adversarial attacks—both untargeted and targeted at specific experts (e.g., $M_3$)—SARD demonstrates significant robustness, maintaining an RMSE of 0.18 across all conditions. In contrast, the baseline's performance degrades substantially, with RMSE values increasing to 0.29 and 0.41 under untargeted and $M_3$-targeted attacks, respectively.

### 6.3. Multi-Task

We evaluate the performance of our robust SARD algorithm against the baseline introduced by Montreuil et al. (2025c) on the Pascal VOC dataset (Everingham et al., 2010), a benchmark for object detection tasks combining both interdependent classification and regression objectives.

**Setting:** We train two Faster R-CNN models (Ren et al., 2015) as experts, each specializing in a distinct subset of the dataset. Expert $M_1$ is trained exclusively on images containing animals, while expert $M_2$ focuses on images with vehicles. Agent costs are defined as $c_0(g(x), z) = \text{mAP}(g(x), z)$ for the model and $c_{j>0}(m_j(x), z) = \text{mAP}(m_j(x), z)$, aligned with (Montreuil et al., 2025c). The primary model and the rejector are implemented as lightweight versions of Faster R-CNN using MobileNet (Howard et al., 2017). We provide detailed performance results, training procedures, and additional experimental details in Appendix E.3.

**Results:** Table 3 presents the performance comparison between SARD and the baseline under clean and adversarial scenarios. Both methods perform comparably in clean conditions, with the baseline achieving a slightly higher

| Baseline | Clean | Untarg. | Targ. $M_1$ | Targ. $M_2$ |
|---|---|---|---|---|
| Montreuil et al. (2025c) | $44.4 \pm 0.4$ | $9.7 \pm 0.1$ | $17.4 \pm 0.2$ | $20.4 \pm 0.2$ |
| Our | $43.9 \pm 0.4$ | $39.0 \pm 0.3$ | $39.7 \pm 0.3$ | $39.5 \pm 0.3$ |

*Table 3.* Performance comparison of SARD with the baseline (Montreuil et al., 2025c) on the Pascal VOC dataset. The table reports mean Average Precision (mAP) under clean and adversarial scenarios.

mAP of 44.4 compared to 43.9 for SARD. However, under adversarial scenarios, the baseline experiences a significant performance drop, with mAP decreasing to 9.7 in untargeted attacks and 17.4 in targeted attacks on $M_1$. In contrast, SARD demonstrates strong robustness, maintaining mAP scores close to the clean setting across all attack types. Specifically, SARD achieves an mAP of 39.0 under untargeted attacks and 39.7 when targeted at $M_1$, highlighting its resilience to adversarial perturbations.

## 7. Conclusion

In this paper, we address the critical and previously underexplored problem of adversarial robustness in two-stage Learning-to-Defer systems. We introduce two novel adversarial attack strategies—untargeted and targeted—that exploit inherent vulnerabilities in existing L2D frameworks. To mitigate these threats, we propose SARD, a robust deferral algorithm that provides theoretical guarantees based on Bayes consistency and $(\mathcal{R}, \mathcal{G})$-consistency. We evaluate our approach across classification, regression, and multi-task scenarios. Our experiments demonstrate the effectiveness of the proposed adversarial attacks in significantly degrading the performance of existing two-stage L2D baselines. In contrast, SARD exhibits strong robustness against these attacks, consistently maintaining high performance.

## Acknowledgment

This research is supported by the National Research Foundation, Singapore under its AI Singapore Programme (AISG Award No: AISG2-PhD-2023-01-041-J) and by A*STAR, and is part of the programme DesCartes which is supported by the National Research Foundation, Prime Minister's Office, Singapore under its Campus for Research Excellence and Technological Enterprise (CREATE) programme.

## Impact Statement

This paper introduces methods to improve the adversarial robustness of two-stage Learning-to-Defer frameworks, which allocate decision-making tasks between AI systems and human experts. The work has the potential to advance the field of Machine Learning, particularly in high-stakes

domains such as healthcare, finance, and safety-critical systems, where robustness and reliability are essential. By mitigating vulnerabilities to adversarial attacks, this research ensures more secure and trustworthy decision-making processes.

The societal implications of this work are largely positive, as it contributes to enhancing the reliability and fairness of AI systems. However, as with any advancement in adversarial robustness, there is a potential for misuse if adversarial strategies are exploited for harmful purposes. While this paper does not directly address these ethical concerns, we encourage further exploration of safeguards and responsible deployment practices in future research.

No immediate or significant ethical risks have been identified in this work, and its societal impacts align with the well-established benefits of improving robustness in Machine Learning systems.

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

# A. Notation and Preliminaries for the Appendices

We summarize the key notations and concepts introduced in the main text:

**Input Space and Outputs:**

- $\mathcal{X}$: Input space for $x \in \mathcal{X}$.

- $\mathcal{Q}$: Latent representation $q \in \mathcal{Q}$.

- $\mathcal{Y} = \{1, \ldots, n\}$: Categorical output space for classification tasks.

- $\mathcal{T} \subseteq \mathbb{R}$: Continuous output space for regression tasks.

- $\mathcal{Z} = \mathcal{X} \times \mathcal{Y} \times \mathcal{T}$: Combined space of inputs and labels.

**Learning-to-Defer Setting:**

- $\mathcal{A} = \{0\} \cup [J]$: Set of agents, where $0$ refers to the primary model $g = (h, f)$, and $J$ denotes the number of experts.

- $m_j(x) = (m_j^h(x), m_j^f(x))$: Predictions by expert $j$, where $m_j^h(x) \in \mathcal{Y}$ is a categorical prediction and $m_j^f(x) \in \mathcal{T}$ is a regression estimate.

- $c_0(g(x), z) = \psi(g(x), z)$: The cost associated to the multi-task model.

- $c_{j>0}(m(x), z) = \psi(m(x), z) + \beta_j$: The cost associated to the expert $j$ with query cost $\beta_j \geq 0$.

- $\psi : \mathcal{Y} \times \mathcal{T} \times \mathcal{Y} \times \mathcal{T} \to \mathbb{R}^+$: Quantify the prediction's quality.

**Hypothesis Sets:**

- $\mathcal{W}$: Set of backbones $w : \mathcal{X} \to \mathcal{Q}$.

- $\mathcal{H}$: Set of classifiers $h : \mathcal{Q} \times \mathcal{Y} \to \mathbb{R}$.

- $\mathcal{F}$: Set of regressors $f : \mathcal{Q} \to \mathcal{T}$.

- $\mathcal{G}$: Single multi-head network $\mathcal{G} = \{g : g(x) = (h \circ w(x), f \circ w(x)) \mid w \in \mathcal{W}, h \in \mathcal{H}, f \in \mathcal{F}\}$.

- $\mathcal{R}$: Set of rejectors $r : \mathcal{X} \to \mathcal{A}$.

**Adversarial Definitions:**

- $x'_j \in B_p(x, \gamma)$: the adversarial input for the agent $j \in \mathcal{A}$ in the $p$-norm ball $B_p(x, \gamma) = \{x'_j \in \mathcal{X} \mid \|x'_j - x\|_p \leq \gamma\}$

- $\widetilde{\ell}_{01}^j(r, x, j) = \sup_{x'_j \in B_p(x,\gamma)} \ell_{01}(r, x'_j, j)$: $j$-th Adversarial multiclass loss.

- $\widetilde{\Phi}_{01}^{\rho,u,j}(r, x, j) = \sup_{x'_j \in B_p(x,\gamma)} \Psi^u \left( \sum_{j' \neq j} \Psi_\rho \big( r(x'_j, j') - r(x'_j, j) \big) \right)$: $j$-th Adversarial margin surrogate losses, providing a differentiable proxy for $\widetilde{\ell}_{01}^j$.

This notation will be consistently used throughout the appendices to ensure clarity and coherence in theoretical and empirical discussions.

# B. Illustration

## B.1. Learning-to-Defer in Inference

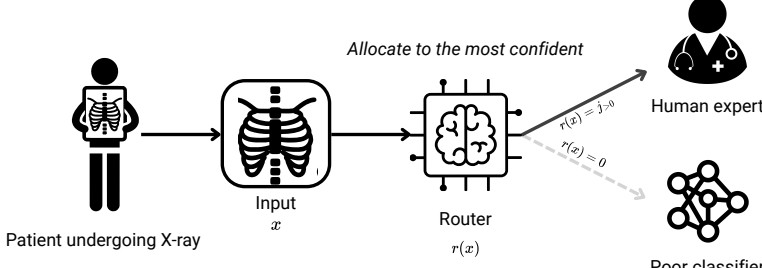

*Figure 1.* Learning-to-Defer in inference. The input $x$ is routed to the most cost-effective agent in the system by the rejector $r \in \mathcal{R}$: in this case, the human expert.

## B.2. Untargeted Attack

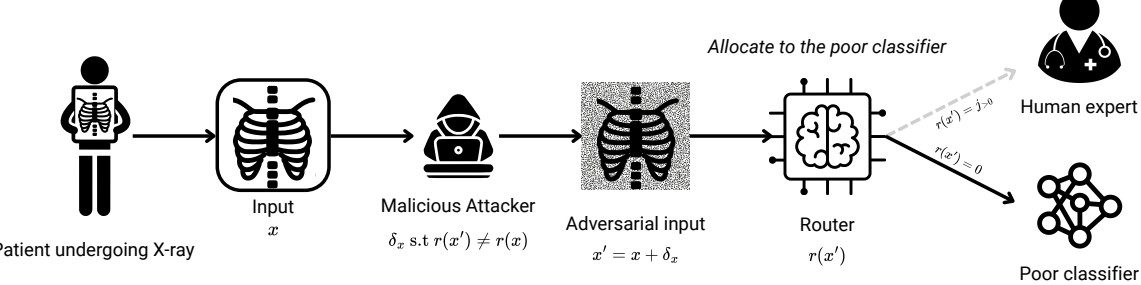

*Figure 2.* Untargeted Attack: The malicious attacker perturbs the input to increase the probability that the query is assigned to a less accurate expert, thereby maximizing classification errors. Rather than targeting a specific expert, the attack injects adversarial noise to disrupt the expert allocation process, leading to erroneous routing and degraded decision-making.

## B.3. Targeted Attack

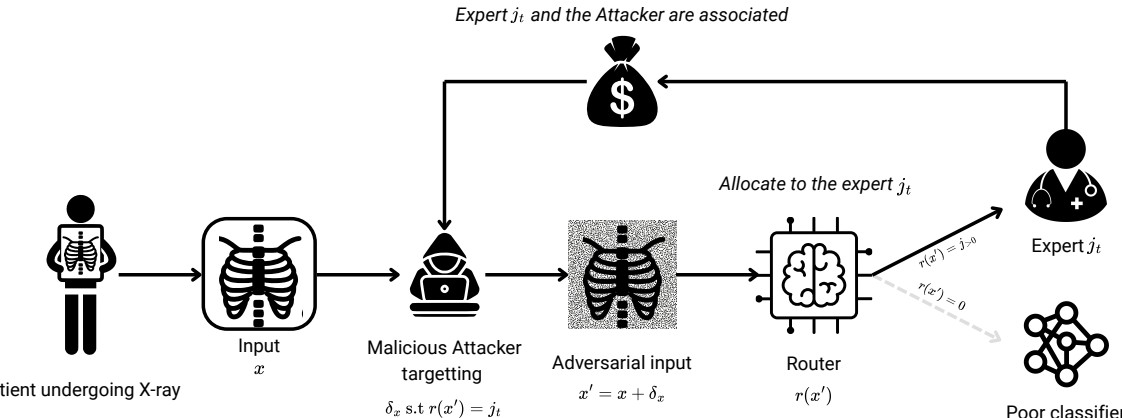

*Figure 3.* Targeted Attack: The malicious attacker perturbs the input to increase the probability that the query is assigned to its associated agent. By manipulating the L2D system to systematically route queries to this associate, the adversary ensures that the associate receives a higher volume of queries, thereby increasing its earnings.

## C. Algorithm

---

**Algorithm 1** SARD Algorithm

---

**Input:** Dataset $\{(x_k, y_k, t_k)\}_{k=1}^K$, multi-task model $g \in \mathcal{G}$, experts $m \in \mathcal{M}$, rejector $r \in \mathcal{R}$, number of epochs EPOCH, batch size BATCH, adversarial parameters $(\rho, \nu)$, regularizer parameter $\eta$, learning rate $\lambda$.
**Initialization:** Initialize rejector parameters $\theta$.
**for** $i = 1$ to EPOCH **do**
    Shuffle dataset $\{(x_k, y_k, t_k)\}_{k=1}^K$.
    **for** each mini-batch $\mathcal{B} \subset \{(x_k, y_k, t_k)\}_{k=1}^K$ of size BATCH **do**
        Extract input-output pairs $z = (x, y, t) \in \mathcal{B}$.
        Query model $g(x)$ and experts $m(x)$.         {Agents have been trained offline and fixed}
        Evaluate costs $c_0(g(x), z)$ and $c_{j>0}(m(x), z)$.         {Compute costs}
        **for** $j = 0$ to $J$ **do**
            Evaluate rejector score $r(x, j)$.         {Rejection score of agent $j$}
            Generate adversarial input $x'_j = x + \delta_j$ with $\delta_j \in B_p(x, \gamma)$.     {$\ell_p$-ball perturbation for agent $j$}
            Run PGD attack on $x'_j$:
            $\sup_{x'_j \in B_p(x, \gamma)} \|\overline{\Delta}_r(x'_j, j) - \overline{\Delta}_r(x, j)\|_2$.         {Smooth robustness evaluation}
            Compute Adversarial Smooth surrogate losses $\widetilde{\Phi}_{01}^{\text{smth},u}(r, x, j)$.
        **end for**
        Compute the regularized empirical risk minimization:
        $\widehat{\mathcal{E}}_{\Phi_{\text{def}}}^{\Omega}(r; \theta) = \frac{1}{\text{BATCH}} \sum_{z \in \mathcal{B}} \left[ \widetilde{\Phi}_{\text{def}}^{\text{smth},u}(r, g, m, z) \right] + \eta \Omega(r)$.
        Update parameters $\theta$:
        $\theta \leftarrow \theta - \lambda \nabla_\theta \widehat{\mathcal{E}}_{\Phi_{\text{def}}}^{\Omega}(r; \theta)$.         {Gradient update}
    **end for**
**end for**
**Return:** trained rejector model $\hat{r}$.

---

## D. Proof Adversarial Robustness in Two-Stage Learning-to-Defer

### D.1. Proof Lemma 5.2

**Lemma 5.2** (Adversarial True Deferral Loss). *Let $x \in \mathcal{X}$ denote the clean input, $c_j$ the cost associated with agent $j \in \mathcal{A}$, and $\tau_j$ the aggregated cost. The adversarial true deferral loss $\widetilde{\ell}_{def}$ is defined as:*

$$\widetilde{\ell}_{def}(r, g, m, z) = \sum_{j=0}^{J} \tau_j(g(x), m(x), z) \widetilde{\ell}_{01}^j(r, x, j)$$

$$+ (1 - J) \sum_{j=0}^{J} c_j(g(x), m_j(x), z).$$

*Proof.* In adversarial training, the objective is to optimize the worst-case scenario of the objective function under adversarial inputs $x' \in B_p(x, \gamma)$. For our case, we start with the standard L2D loss for the two-stage setting (Mao et al., 2024d; Montreuil et al., 2025c):

$$
\begin{aligned}
\ell_{\text{def}}(r, g, m, z) &= \sum_{j=0}^{J} c_j(g(x), m_j(x), z) 1_{r(x)=j} \\
&= \sum_{j=0}^{J} \tau_j(g(x), m(x), z) 1_{r(x) \neq j} + (1 - J) \sum_{j=0}^{J} c_j(g(x), m_j(x), z)
\end{aligned}
\tag{5}
$$

using

$$\tau_j(g(x), m(x), z) = \begin{cases} \sum_{i=1}^{J} c_i(m_i(x), z) & \text{if } j = 0 \\ c_0(g(x), z) + \sum_{i=1}^{J} c_i(m_i(x), z) 1_{i \neq j} & \text{otherwise} \end{cases} \tag{6}$$

Next, we derive an upper bound for Equation (5) by considering the supremum over all adversarial perturbations $x' \in B_p(x, \gamma)$, under the fact that the attack is solely on the rejector $r \in \mathcal{R}$:

$$\ell_{\text{def}}(r, g, m, z) \leq \sup_{x' \in B_p(x, \gamma)} \left( \sum_{j=0}^{J} \tau_j(g(x), m(x), z) 1_{r(x') \neq j} \right) + (1 - J) \sum_{j=0}^{J} c_j(g(x), m_j(x), z) \tag{7}$$

However, the formulation in Equation (7) does not fully capture the worst-case scenario in L2D. Specifically, this formulation might not result in a robust approach, as it does not account for the adversarial input $x'_j \in B_p(x, \gamma)$ that maximizes the loss for every agent $j \in \mathcal{A}$. Incorporating this worst-case scenario, we obtain:

$$\ell_{\text{def}}(r, g, m, z) \leq \sum_{j=0}^{J} \tau_j(g(x), m(x), z) \sup_{x'_j \in B_p(x, \gamma)} 1_{r(x'_j) \neq j} + (1 - J) \sum_{j=0}^{J} c_j(g(x), m_j(x), z) \tag{8}$$

Thus, formulating with the margin loss $\rho_r(x, j) = r(x, j) - \max_{j' \neq j} r(x, j')$, leads to the desired result:

$$\begin{aligned} \widetilde{\ell}_{\text{def}}(r, g, m, z) &= \sum_{j=0}^{J} \tau_j(g(x), m(x), z) \sup_{x'_j \in B_p(x, \gamma)} 1_{\rho_r(x'_j, j) \leq 0} + (1 - J) \sum_{j=0}^{J} c_j(g(x), m_j(x), z) \\ &= \sum_{j=0}^{J} \tau_j(g(x), m(x), z) \widetilde{\ell}_{01}^j(r, x, j) + (1 - J) \sum_{j=0}^{J} c_j(g(x), m_j(x), z) \end{aligned} \tag{9}$$

with $\widetilde{\ell}_{01}^j(r, x, j) = \sup_{x'_j \in B_p(x, \gamma)} 1_{\rho_r(x'_j, j) \leq 0}$ ☐

### D.2. Proof Lemma 5.5

**Lemma 5.5** (Adversarial Margin Deferral Surrogate Losses)**.** *Let $x \in \mathcal{X}$ denote the clean input and $\tau_j$ the aggregated cost. The adversarial margin deferral surrogate losses $\widetilde{\Phi}_{def}^{\rho, u}$ are then defined as:*

$$\widetilde{\Phi}_{def}^{\rho, u}(r, g, m, z) = \sum_{j=0}^{J} \tau_j(g(x), m(x), z) \widetilde{\Phi}_{01}^{\rho, u, j}(r, x, j).$$

*Proof.* Referring to adversarial true deferral loss defined in Lemma 5.2, we have:

$$\begin{aligned} \widetilde{\ell}_{\text{def}}(r, g, m, z) &= \sum_{j=0}^{J} \tau_j(g(x), m(x), z) \sup_{x'_j \in B_p(x, \gamma)} 1_{\rho_r(x'_j, j) \leq 0} + (1 - J) \sum_{j=0}^{J} c_j(g(x), m_j(x), z) \\ &= \sum_{j=0}^{J} \tau_j(g(x), m(x), z) \widetilde{\ell}_{01}^j(r, x, j) + (1 - J) \sum_{j=0}^{J} c_j(g(x), m_j(x), z) \end{aligned}$$

By definition, $\widetilde{\Phi}_{01}^{\rho, u, j}$ upper bounds the $j$-th adversarial classification loss $\widetilde{\ell}_{01}^j$, leading to:

$$\widetilde{\ell}_{\text{def}}(r, g, m, z) \leq \sum_{j=0}^{J} \tau_j(g(x), m(x), z) \widetilde{\Phi}_{01}^{\rho, u, j}(r, x, j) + (1 - J) \sum_{j=0}^{J} c_j(g(x), m_j(x), z) \tag{10}$$

Then, dropping the term that does not depend on $r \in \mathcal{R}$, leads to the desired formulation:

$$\widetilde{\Phi}_{\text{def}}^{\rho, u}(r, g, m, z) = \sum_{j=0}^{J} \tau_j(g(x), m(x), z) \widetilde{\Phi}_{01}^{\rho, u, j}(r, x, j) \tag{11}$$

☐

## D.3. Proof Lemma 5.6

**Lemma 5.6** (Smooth Adversarial Surrogate Losses). *Let $x \in \mathcal{X}$ denote the clean input and hyperparameters $\rho > 0$ and $\nu > 0$. The smooth adversarial surrogate losses are defined as:*

$$
\widetilde{\Phi}_{01}^{smth,u}(r,x,j) = \Phi_{01}^u(\frac{r}{\rho},x,j)
$$
$$
+ \nu \sup_{x'_j \in B_p(x,\gamma)} \|\overline{\Delta}_r(x'_j,j) - \overline{\Delta}_r(x,j)\|_2.
$$

*Proof.* Let $x \in \mathcal{X}$ denote an input and $x'_j \in B_p(x,\gamma)$ an adversarially perturbed input within an $\ell_p$-norm ball of radius $\gamma$ for each agent. Let $r \in \mathcal{R}$ be a rejector. We now define the composite-sum $\rho$-margin losses for both clean and adversarial scenarios:

$$
\Phi_{01}^{\rho,u}(r,x,j) = \Psi^u\left(\sum_{j' \neq j} \Psi_\rho\big(r(x,j') - r(x,j)\big)\right)
$$
$$
\widetilde{\Phi}_{01}^{\rho,u,j}(r,x,j) = \sup_{x'_j \in B_p(x,\gamma)} \Psi^u\left(\sum_{j' \neq j} \Psi_\rho\big(r(x'_j,j') - r(x'_j,j)\big)\right)
$$

$$(12)$$

where $\Psi_e(v) = \exp(-v)$. For $u > 0$, the transformation $\Psi^u$ is defined as:

$$
\Psi^{u=1}(v) = \log(1+v), \quad \Psi^{u \neq 1}(v) = \frac{1}{1-u}\left[(1-v)^{1-u} - 1\right]
$$

It follows that for all $u > 0$ and $v \geq 0$, we have $\left|\frac{\partial \Psi^u}{\partial v}(v)\right| = \frac{1}{(1+v)^u} \leq 1$ ensuring that $\Psi^u$ is 1-Lipschitz over $\mathbb{R}^+$ (Mao et al., 2023b).

Define $\Delta_r(x,j,j') = r(x,j) - r(x,j')$ and let $\overline{\Delta}_r(x,j)$ denote the $J$-dimensional vector:

$$
\overline{\Delta}_r(x,j) = \big(\Delta_r(x,j,0),\ldots,\Delta_r(x,j,j-1),\Delta_r(x,j,j+1),\ldots,\Delta_r(x,j,J)\big)
$$

For any $u > 0$, with $\Psi^u$ non-decreasing and 1-Lipschitz:

$$
\widetilde{\Phi}_{01}^{\rho,j}(r,x,j) \leq \Phi_{01}^{\rho,u}(r,x,j) + \sup_{x'_j \in B_p(x,\gamma)} \sum_{j' \neq j}\left(\Psi_\rho\big(-\Delta_r(x'_j,j,j')\big) - \Psi_\rho\big(-\Delta_r(x,j,j')\big)\right)
$$

$$(13)$$

Since $\Psi_\rho(z)$ is $\frac{1}{\rho}$-Lipschitz, by the Cauchy-Schwarz inequality and for $\nu \geq \frac{\sqrt{n-1}}{\rho} \geq \frac{1}{\rho}$:

$$
\widetilde{\Phi}_{01}^{\rho,u,j}(r,x,j) \leq \Phi_{01}^{\rho,u}(r,x,j) + \nu \sup_{x'_j \in B_p(x,\gamma)} \|\overline{\Delta}_r(x'_j,j) - \overline{\Delta}_r(x,j)\|_2
$$

$$(14)$$

Using $\Phi_{01}^u(r,x,y) = \Psi^u\big(\sum_{y' \neq y} \Psi_e(r(x,y) - r(x,y'))\big)$ with $\Psi_e(v) = \exp(-v)$ and the fact that $\Psi_e(v/\rho) \geq \Psi_\rho(v)$, we obtain:

$$
\widetilde{\Phi}_{01}^{\rho,u,j}(r,x,j) \leq \Phi_{01}^u\left(\frac{r}{\rho},x,j\right) + \nu \sup_{x'_j \in B_p(x,\gamma)} \|\overline{\Delta}_r(x'_j,j) - \overline{\Delta}_r(x,j)\|_2
$$

$$(15)$$

Finally, we have the desired smooth surrogate losses upper-bounding $\widetilde{\Phi}_{01}^{\text{smth},u} \geq \widetilde{\Phi}_{01}^{\rho,u,j}$:

$$
\widetilde{\Phi}_{01}^{\text{smth},u}(r,x,j) = \Phi_{01}^u\left(\frac{r}{\rho},x,j\right) + \nu \sup_{x'_j \in B_p(x,\gamma)} \|\overline{\Delta}_r(x'_j,j) - \overline{\Delta}_r(x,j)\|_2
$$

$$(16)$$

$\square$

## D.4. Proof Lemma 5.7

**Lemma 5.7** (SAD: **S**mooth **A**dversarial **D**eferral Surrogate Losses)**.** *Let $x \in \mathcal{X}$ denote the clean input and $\tau_j$ the aggregated cost. Then, the smooth adversarial surrogate family (SAD) $\widetilde{\Phi}_{def}^{smth,u}$ is defined as:*

$$\widetilde{\Phi}_{def}^{smth,u}(r, g, m, z) = \sum_{j=0}^{J} \tau_j(g(x), m(x), z) \widetilde{\Phi}_{01}^{smth,u}(r, x, j)$$

*Proof.* Using Lemma 5.5, we have:

$$\widetilde{\Phi}_{def}^{\rho,u}(r, g, m, z) = \sum_{j=0}^{J} \tau_j(g(x), m(x), z) \widetilde{\Phi}_{01}^{\rho,j}(r, x, j) \tag{17}$$

Since $\widetilde{\Phi}_{01}^{\rho,u,j} \leq \widetilde{\Phi}_{01}^{smth,u}$ by Lemma 5.6, we obtain:

$$\widetilde{\Phi}_{def}^{smth,u}(r, g, m, z) = \sum_{j=0}^{J} \tau_j(g(x), m(x), z) \widetilde{\Phi}_{01}^{smth,u}(r, x, j) \tag{18}$$

$\square$

## D.5. Proof Lemma 5.9

**Lemma 5.9** ($\mathcal{R}$-consistency bounds for $\widetilde{\Phi}_{01}^{\rho,u,j}$)**.** *Assume $\mathcal{R}$ is symmetric and locally $\rho$-consistent. Then, for the agent set $\mathcal{A}$, any hypothesis $r \in \mathcal{R}$, and any distribution $\mathcal{P}$ with probabilities $p = (p_0, \ldots, p_J) \in \Delta^{|\mathcal{A}|}$, the following inequality holds:*

$$\sum_{j \in \mathcal{A}} p_j \widetilde{\ell}_{01}^j(r, x, j) - \inf_{r \in \mathcal{R}} \sum_{j \in \mathcal{A}} p_j \widetilde{\ell}_{01}^j(r, x, j) \leq$$
$$\Psi^u(1) \Big( \sum_{j \in \mathcal{A}} p_j \widetilde{\Phi}_{01}^{\rho,u,j}(r, x, j) - \inf_{r \in \mathcal{R}} \sum_{j \in \mathcal{A}} p_j \widetilde{\Phi}_{01}^{\rho,u,j}(r, x, j) \Big).$$

*Proof.* We define the margin as $\rho_r(x, j) = r(x, j) - \max_{j' \neq j} r(x, j')$, which quantifies the difference between the score of the $j$-th dimension and the highest score among all other dimensions. Starting from this, we can define a space $\overline{\mathcal{R}}_\gamma(x) = \{r \in \mathcal{R} : \inf_{x' \in B_p(x,\gamma)} \rho_r(x', r(x)) > 0\}$ for $B_p(x, \gamma) = \{x' \mid \|x' - x\|_p \leq \gamma\}$ representing hypothesis that correctly classifies the adversarial input. By construction, we have that $x'_j \in B_p(x, \gamma)$.

In the following, we will make use of several notations. Let $p(x) = (p(x, 0), \ldots, p(x, J))$ denote the probability distribution over $\mathcal{A}$ at point $x \in \mathcal{X}$. We sort these probabilities $\{p(x, j) : j \in \mathcal{A}\}$ in increasing order $p_{[0]}(x) \leq p_{[1]}(x) \leq \cdots \leq p_{[J]}(x)$. Let $\mathcal{R}$ be a hypothesis class for the rejector $r \in \mathcal{R}$ with $r : \mathcal{X} \times \mathcal{A} \to \mathbb{R}$. We assume this hypothesis class to be *symmetric* implying that for any permutation $\pi$ of $\mathcal{A}$ and any $r \in \mathcal{R}$, the function $r^\pi$ defined by $r^\pi(x, j) = r(x, \pi(j))$ is also in $\mathcal{R}$ for $j \in \mathcal{A}$. We similarly have $r \in \mathcal{R}$, such that $r(x, \{0\}_x^r), r(x, \{1\}_x^r), \cdots, r(x, \{J\}_x^r)$ sorting the scores $\{r(x, j) : j \in \mathcal{A}\}$ in increasing order.

For $\mathcal{R}$ symmetric and locally $\rho$-consistent, there exists $r^* \in \mathcal{R}$ with the same ordering of the $j \in \mathcal{A}$, regardless of any $x'_j \in B_p(x, \gamma)$. This implies $\inf_{x'_j \in B_p(x,\gamma)} |r^*(x'_j, q) - r^*(x'_j, q')| \geq \rho$ for $\forall q' \neq q \in \mathcal{A}$. Using the symmetry of $\mathcal{R}$, we can find a $r^*$ with the same ordering of $j \in \mathcal{A}$, i.e. $p(x, \{k\}_x^{r^*}) = p_{[k]}(x)$ for any $k \in \mathcal{A}$:

$$\forall j \in \mathcal{A}, \quad r^*(x'_j, \{0\}_{x'_j}^{r^*}) \leq r^*(x'_j, \{1\}_{x'_j}^{r^*}) \leq \cdots \leq r^*(x'_j, \{J\}_{x'_j}^{r^*}) \tag{19}$$

We introduce a new notation $\xi'_k = x'_{\{k\}}$ corresponding to the $k$-th ordered adversarial input. For instance, if we have an ordered list $\{r^*(x'_2, 2), r^*(x'_0, 0), r^*(x'_1, 1)\}$, using the notation we have $\{r^*(\xi'_0, \{0\}_{\xi'_0}^{r^*}), r^*(\xi'_1, \{1\}_{\xi'_1}^{r^*}), r^*(\xi'_2, \{2\}_{\xi'_2}^{r^*})\}$.

We define a conditional risk $\mathcal{C}_{\widetilde{\Phi}_{01}^{\rho,u,j}}$ parameterized by the probability $p_j \in \Delta^{|\mathcal{A}|}$ along with its optimum:

$$\mathcal{C}_{\widetilde{\Phi}_{01}^{\rho,u,j}}(r,x) = \sum_{j \in \mathcal{A}} p_j \widetilde{\Phi}_{01}^{\rho,u,j}(r,x,j)$$

$$\mathcal{C}_{\widetilde{\Phi}_{01}^{\rho,u,j}}^*(\mathcal{R},x) = \inf_{r \in \mathcal{R}} \sum_{j \in \mathcal{A}} p_j \widetilde{\Phi}_{01}^{\rho,u,j}(r,x,j)$$

(20)

The optimum $\mathcal{C}_{\widetilde{\Phi}_{01}^{\rho,u,j}}^*$ is challenging to characterize directly. To address this, we instead derive an upper bound by analyzing $\mathcal{C}_{\widetilde{\Phi}_{01}^{\rho,u,j}}(r^*,x)$. In what follows, the mapping from $j$ to $i$ is defined based on the rank of $p(x,j)$ within the sorted list $\{p_{[i]}(x)\}$.

$$\mathcal{C}_{\widetilde{\Phi}_{01}^{\rho,u,j}}^*(\mathcal{R},x) \leq \mathcal{C}_{\widetilde{\Phi}_{01}^{\rho,u,j}}(r^*,x)$$

$$= \sum_{j \in \mathcal{A}} p(x,j) \sup_{x_j' \in B_p(x,\gamma)} \Psi^u\left( \sum_{\substack{j' \in \mathcal{A} \\ j' \neq j}} \Psi_\rho\left( r^*(x_j',j) - r^*(x_j',j') \right) \right)$$

$$= \sum_{i=0}^{J} \sup_{\xi_i' \in B_p(x,\gamma)} p(x,\{i\}_{\xi_i'}^{r^*}) \Psi^u\left( \sum_{j=0}^{i-1} \Psi_\rho\left( r^*(\xi_i',\{i\}_{\xi_i'}^{r^*}) - r^*(\xi_i',\{j\}_{\xi_i'}^{r^*}) \right) \right.$$

$$\left. + \sum_{j=i+1}^{J} \Psi_\rho\left( r^*(\xi_i',\{i\}_{\xi_i'}^{r^*}) - r^*(\xi_i',\{j\}_{\xi_i'}^{r^*}) \right) \right)$$

$$= \sum_{i=0}^{J} \sup_{\xi_i' \in B_p(x,\gamma)} p(x,\{i\}_{\xi_i'}^{r^*}) \Psi^u\left( \sum_{j=0}^{i-1} \Psi_\rho\left( r^*(\xi_i',\{i\}_{\xi_i'}^{r^*}) - r^*(\xi_i',\{j\}_{\xi_i'}^{r^*}) \right) + J - i \right) \quad \left( \Psi_\rho(t) = 1, \forall t \leq 0 \right)$$

$$= \sum_{i=0}^{J} \sup_{\xi_i' \in B_p(x,\gamma)} p(x,\{i\}_{\xi_i'}^{r^*}) \Psi^u(J-i) \quad \left( \Psi_\rho(v) = 0, \forall v \geq \rho \text{ and } \inf_{x_j' \in B_p(x,\gamma)} |r^*(x_j',q) - r^*(x_j',q')| \geq \rho \right)$$

$$= \sum_{i=0}^{J} p_{[i]}(x) \Psi^u(J-i) \quad \left( r^* \text{ and } p(x) \text{ same ordering of } j \in \mathcal{A} \right)$$

(21)

Then, assuming $\overline{\mathcal{R}}_\gamma(x) \neq \varnothing$ and $\mathcal{R}$ symmetric, we have:

$$\Delta\mathcal{C}_{\widetilde{\Phi}_{01}^{\rho,u,j}}(r,x) = \mathcal{C}_{\widetilde{\Phi}_{01}^{\rho,u,j}}(r,x) - \mathcal{C}_{\widetilde{\Phi}_{01}^{\rho,u,j}}^*(\mathcal{R},x) \geq \mathcal{C}_{\widetilde{\Phi}_{01}^{\rho,u,j}}(r,x) - \mathcal{C}_{\widetilde{\Phi}_{01}^{\rho,u,j}}(r^*,x)$$

$$\geq \sum_{i=0}^{J} \sup_{\xi_i' \in B_p(x,\gamma)} p(x,\{i\}_{\xi_i'}^{r}) \Psi^u\left( \sum_{j=0}^{i-1} \Psi_\rho\left( r(\xi_i',\{i\}_{\xi_i'}) - r(\xi_i',\{j\}_{\xi_i'}) \right) + J - i \right)$$

$$- \left( \sum_{i=0}^{J} p_{[i]}(x) \Psi^u(J-i) \right)$$

(22)

Then, for $\Psi_\rho$ non negative, $\Psi_\rho(v) = 1$ for $v \leq 0$, and $\Psi^u$ non-decreasing, we have that:

$$\Delta\mathcal{C}_{\widetilde{\Phi}_{01}^{\rho,u,j}}(r,x) \geq \Psi^u(1) p(x,r(x)) 1_{r \notin \overline{\mathcal{R}}_\gamma(x)} + \sum_{i=0}^{J} \sup_{\xi_i' \in B_p(x,\gamma)} p(x,\{i\}_{\xi_i'}^{r}) \Psi^u(J-i) - \left( \sum_{i=0}^{J} p_{[i]}(x) \Psi^u(J-i) \right)$$

$$\geq \Psi^u(1) p(x,r(x)) 1_{r \notin \overline{\mathcal{R}}_\gamma(x)} - \sum_{i=0}^{J} p_{[i]}(x) \Psi^u(J-i) + \sum_{i=0}^{J} p(x,\{i\}_x^{r}) \Psi^u(J-i)$$

$$(\sup_{\xi_i' \in B_p(x,\gamma)} p(x,\{i\}_{\xi_i'}^{r}) \geq p(x,\{i\}_x^{r})$$

Then RHS is equal to the above,

$$\Delta \mathcal{C}_{\widetilde{\Phi}_{01}^{\rho,u,j}}(r,x) \geq \Psi^u(1)p(x,r(x))1_{r \notin \overline{\mathcal{R}}_\gamma(x)} + \Psi^u(1)\left(\max_{j \in \mathcal{A}} p(x,j) - p(x,r(x))\right) + \begin{bmatrix} \Psi^u(1) \\ \Psi^u(1) \\ \Psi^u(2) \\ \vdots \\ \Psi^u(J) \end{bmatrix} \cdot \begin{bmatrix} p(x,\{J\}_x^r) \\ p(x,\{J-1\}_x^r) \\ p(x,\{J-2\}_x^r) \\ \vdots \\ p(x,\{0\}_x^r) \end{bmatrix}$$

$$\tag{23}$$

$$- \begin{bmatrix} \Psi^u(1) \\ \Psi^u(1) \\ \Psi^u(2) \\ \vdots \\ \Psi^u(J) \end{bmatrix} \cdot \begin{bmatrix} p_{[J]}(x) \\ p_{[J-1]}(x) \\ p_{[J-2]}(x) \\ \vdots \\ p_{[0]}(x) \end{bmatrix} \tag{24}$$

Rearranging terms for $\Psi^u(1) \leq \Psi^u(1) \leq \Psi^u(2) \leq \cdots \leq \Psi^u(J)$ and similarly for probabilities $p_{[J]}(x) \geq \cdots \geq p_{[0]}(x)$, leads to:

$$\Delta \mathcal{C}_{\widetilde{\Phi}_{01}^{\rho,u,j}}(r,x) \geq \Psi^u(1)p(x,r(x))1_{r \notin \overline{\mathcal{R}}_\gamma(x)} + \Psi^u(1)\left(\max_{j \in \mathcal{A}} p(x,j) - p(x,r(x))\right)$$

$$= \Psi^u(1)\left(\max_{j \in \mathcal{A}} p(x,j) - p(x,r(x))1_{r \in \overline{\mathcal{R}}_\gamma(x)}\right)$$

$$\tag{25}$$

for any $r \in \mathcal{R}$, we have:

$$\Delta \mathcal{C}_{\widetilde{\ell}_{01}^j}(r,x) = \mathcal{C}_{\widetilde{\ell}_{01}^j}(r,x) - \mathcal{C}_{\widetilde{\ell}_{01}^j}^B(\mathcal{R},x)$$

$$= \sum_{j \in \mathcal{A}} p(x,j) \sup_{x'_j \in B_p(x,\gamma)} 1_{\rho_r(x'_j,j) \leq 0} - \inf_{r \in \mathcal{R}} \sum_{j \in \mathcal{A}} p(x,j) \sup_{x'_j \in B_p(x,\gamma)} 1_{\rho_r(x'_j,j) \leq 0}$$

$$= (1 - p(x,r(x)))1_{r \in \overline{\mathcal{R}}_\gamma(x)} + 1_{r \notin \overline{\mathcal{R}}_\gamma(x)} - \inf_{r \in \mathcal{R}} \left[(1 - p(x,r(x)))1_{r \in \overline{\mathcal{R}}_\gamma(x)} + 1_{r \notin \overline{\mathcal{R}}_\gamma(x)}\right]$$

$$= (1 - p(x,r(x)))1_{r \in \overline{\mathcal{R}}_\gamma(x)} + 1_{r \notin \overline{\mathcal{R}}_\gamma(x)} - \left(1 - \max_{j \in \mathcal{A}} p(x,j)\right) \quad (\mathcal{R} \text{ is symmetric and } \overline{\mathcal{R}}_\gamma(x) \neq \varnothing)$$

$$= \max_{j \in \mathcal{A}} p(x,j) - p(x,r(x))1_{r \in \overline{\mathcal{R}}_\gamma(x)} \tag{26}$$

We therefore have proven that:

$$\Delta \mathcal{C}_{\widetilde{\ell}_{01}^j}(r,x) \leq \Psi^u(1)\left(\Delta \mathcal{C}_{\widetilde{\Phi}_{01}^{\rho,u,j}}(r,x)\right)$$

$$\sum_{j \in \mathcal{A}} p_j \widetilde{\ell}_{01}^j(r,x,j) - \inf_{r \in \mathcal{R}} \sum_{j \in \mathcal{A}} p_j \widetilde{\ell}_{01}^j(r,x,j) \leq \Psi^u(1)\left(\sum_{j \in \mathcal{A}} p_j \widetilde{\Phi}_{01}^{\rho,u,j}(r,x,j) - \inf_{r \in \mathcal{R}} \sum_{j \in \mathcal{A}} p_j \widetilde{\Phi}_{01}^{\rho,u,j}(r,x,j)\right) \tag{27}$$

$$\square$$

### D.6. Proof Theorem 5.10

**Theorem 5.10** (($\mathcal{R},\mathcal{G}$)-consistency bounds of $\widetilde{\Phi}_{\text{def}}^{\rho,u}$)**.** *Let $\mathcal{R}$ be symmetric and locally $\rho$-consistent. Then, for the agent set $\mathcal{A}$, any hypothesis $r \in \mathcal{R}$, and any distribution $\mathcal{D}$, the following holds for a multi-task model $g \in \mathcal{G}$:*

$$\mathcal{E}_{\widetilde{\ell}_{def}}(r,g) - \mathcal{E}_{\widetilde{\ell}_{def}}^B(\mathcal{R},\mathcal{G}) + \mathcal{U}_{\widetilde{\ell}_{def}}(\mathcal{R},\mathcal{G}) \leq$$

$$\Psi^u(1)\left(\mathcal{E}_{\widetilde{\Phi}_{def}^{\rho,u}}(r) - \mathcal{E}_{\widetilde{\Phi}_{def}^{\rho,u}}^*(\mathcal{R}) + \mathcal{U}_{\widetilde{\Phi}_{def}^{\rho,u}}(\mathcal{R})\right)$$

$$+ \mathcal{E}_{c_0}(g) - \mathcal{E}_{c_0}^B(\mathcal{G}) + \mathcal{U}_{c_0}(\mathcal{G}).$$

*Proof.* Using Lemma 5.5, we have:

$$\widetilde{\Phi}_{\text{def}}^{\rho,u}(r,g,m,z) = \sum_{j=0}^{J} \tau_j(g(x),m(x),z)\widetilde{\Phi}_{01}^{\rho,j}(r,x,j) \tag{28}$$

We define several important notations. For a quantity $\omega \in \mathbb{R}$, we note $\overline{\omega}(g(x),x) = \mathbb{E}_{y,t|x}[\omega(g,z=(x,y,t))]$, an optimum $\omega^*(z) = \inf_{g\in\mathcal{G}}[\omega(g,z)]$, and the combination $\overline{w}^*(x) = \inf_{g\in\mathcal{G}} \mathbb{E}_{y,t|x}[w(g,z)]$:

$$c_j^*(m_j(x),z) = \begin{cases} c_0^*(z) = \inf_{g\in\mathcal{G}}[c_0(g(x),z)] & \text{if } j=0 \\ c_j(m_j(x),z) & \text{otherwise} \end{cases} \tag{29}$$

Furthermore,

$$\tau_j^*(m(x),z) = \begin{cases} \tau_0(m(x),z) = \sum_{k=1}^{J} c_k(m_k,z) & \text{if } j=0 \\ \inf_{g\in\mathcal{G}}[\tau_j(g(x),m(x),z)] = c_0^*(z) + \sum_{k=1}^{J} c_k(m_k(x),z)1_{k\neq j} & \text{otherwise} \end{cases} \tag{30}$$

Next, we define the conditional risk $\mathcal{C}_{\widetilde{\ell}_{\text{def}}}$ associated to the adversarial true deferral loss.

$$\begin{aligned}
\mathcal{C}_{\widetilde{\ell}_{\text{def}}}(r,g,x) &= \mathbb{E}_{y,t|x}\left[\sum_{j=0}^{J}\tau_j(g(x),m(x),z)\widetilde{\ell}_{01}^j(r,x,j) + (1-J)\sum_{j=0}^{J}c_j(g(x),m_j(x),z)\right] \\
&= \sum_{j=0}^{J}\overline{\tau}_j(g(x),m(x),x)\widetilde{\ell}_{01}^j(r,x,j) + (1-J)\sum_{j=0}^{J}\overline{c}_j(g(x),m_j(x),x)
\end{aligned} \tag{31}$$

Now, we assume $r \in \mathcal{R}$ symmetric and define the space $\overline{\mathcal{R}}_\gamma(x) = \{r \in \mathcal{R} : \inf_{x'\in B_p(x,\gamma)} \rho_r(x',r(x)) > 0\}$. Assuming $\overline{\mathcal{R}}_\gamma(x) \neq \varnothing$, it follows:

$$\mathcal{C}_{\widetilde{\ell}_{\text{def}}}(r,g,x) = \sum_{j=0}^{J}\left(\overline{\tau}_j(g(x),m(x),x)[1_{r(x)\neq j}1_{r\in\overline{\mathcal{R}}_\gamma(x)} + 1_{r\notin\overline{\mathcal{R}}_\gamma(x)}]\right) + (1-J)\sum_{j=0}^{J}\overline{c}_j(g(x),m_j(x),x) \tag{32}$$

Intuitively, if $r \notin \mathcal{R}_\gamma(x)$, this means that there is no $r$ that correctly classifies $x' \in B_p(x,\gamma)$ inducing an error of $1$. It follows at the optimum:

$$\begin{aligned}
\mathcal{C}_{\widetilde{\ell}_{\text{def}}}^B(\mathcal{R},\mathcal{G},x) &= \inf_{g\in\mathcal{G},r\in\mathcal{R}}\left[\sum_{j=0}^{J}\left(\overline{\tau}_j(g(x),m(x),x)[1_{r(x)\neq j}1_{r\in\overline{\mathcal{R}}_\gamma(x)} + 1_{r\notin\overline{\mathcal{R}}_\gamma(x)}]\right) + (1-J)\sum_{j=0}^{J}\overline{c}_j(g(x),m_j(x),x)\right] \\
&= \inf_{r\in\mathcal{R}}\left[\sum_{j=0}^{J}\left(\overline{\tau}_j^*(m(x),x)[1_{r(x)\neq j}1_{r\in\overline{\mathcal{R}}_\gamma(x)} + 1_{r\notin\overline{\mathcal{R}}_\gamma(x)}]\right)\right] + (1-J)\sum_{j=0}^{J}\overline{c}_j^*(m_j(x),x) \\
&= \inf_{r\in\mathcal{R}}\sum_{j=0}^{J}\left(\overline{\tau}_j^*(m(x),x)1_{r(x)\neq j}\right) + (1-J)\sum_{j=0}^{J}\overline{c}_j^*(m_j(x),x) \quad (\overline{\mathcal{R}}_\gamma(x)\neq\varnothing, \text{ then } \exists r\in\overline{\mathcal{R}}_\gamma(x)) \\
&= \sum_{j=0}^{J}\overline{\tau}_j^*(m(x),x)(1-\sup_{r\in\mathcal{R}}1_{r(x)=j}) + (1-J)\sum_{j=0}^{J}\overline{c}_j^*(m_j(x),x) \\
&= \sum_{j=0}^{J}\overline{\tau}_j^*(m(x),x) - \max_{j\in\mathcal{A}}\overline{\tau}_j^*(m(x),x) + (1-J)\sum_{j=0}^{J}\overline{c}_j^*(m_j(x),x)
\end{aligned} \tag{33}$$

We can still work on making the last expression simpler:

$$
\begin{aligned}
\sum_{j=0}^{J} \overline{\tau}_j^*(m(x), x) &= \sum_{j=1}^{J} \overline{c}_j(m_j(x), x) + \sum_{j=1}^{J} \left( \overline{c}_0^*(x) + \sum_{k=1}^{J} \overline{c}_k(m_k(x), x) 1_{k \neq j} \right) \\
&= J \overline{c}_0^*(x) + \sum_{j=1}^{J} \left( \overline{c}_j(m_j(x), x) + \sum_{k=1}^{J} \overline{c}_k(m_k(x), x) 1_{k \neq j} \right) \\
&= J \overline{c}_0^*(x) + \sum_{j=1}^{J} \left( \overline{c}_j(m_j(x), x) + \sum_{k=1}^{J} \overline{c}_k(m_k(x), x)(1 - 1_{k=j}) \right) \\
&= J \overline{c}_0^*(x) + \sum_{j=1}^{J} \sum_{k=1}^{J} \overline{c}_k(m_k(x), x) \\
&= J \left( \overline{c}_0^*(x) + \sum_{j=1}^{J} \overline{c}_j(m_j(x), x) \right)
\end{aligned}
\tag{34}
$$

Then, reinjecting (34) in (33) gives:

$$
\mathcal{C}_{\widetilde{\ell}_{\mathrm{def}}}^B(\mathcal{R}, \mathcal{G}, x) = \sum_{j=0}^{J} \overline{c}_j^*(m_j(x), x) - \max_{j \in \mathcal{A}} \overline{\tau}_j^*(m(x), x)
\tag{35}
$$

if $j = 0$:

$$
\begin{aligned}
\mathcal{C}_{\widetilde{\ell}_{\mathrm{def}}}^B(\mathcal{R}, \mathcal{G}, x) &= \sum_{j=0}^{J} \overline{c}_j^*(m_j(x), x) - \overline{\tau}_0(m(x), x) \\
&= \sum_{j=0}^{J} \overline{c}_j^*(m_j(x), x) - \sum_{j=1}^{J} \overline{c}_j(m_j(x), x) \\
&= \overline{c}_0^*(x)
\end{aligned}
\tag{36}
$$

if $j \neq 0$:

$$
\begin{aligned}
\mathcal{C}_{\widetilde{\ell}_{\mathrm{def}}}^B(\mathcal{R}, \mathcal{G}, x) &= \sum_{j=0}^{J} \overline{c}_j^*(m_j(x), x) - \overline{\tau}_{j>0}^*(m(x), x) \\
&= \sum_{j=0}^{J} \overline{c}_j^*(m_j(x), x) - \left( \overline{c}_0^*(x) + \sum_{k=1}^{J} \overline{c}_k(m_k(x), x) 1_{k \neq 1} \right) \\
&= \overline{c}_{j>0}(m_j(x), x)
\end{aligned}
\tag{37}
$$

Therefore, it can be reduced to:

$$
\mathcal{C}_{\widetilde{\ell}_{\mathrm{def}}}^B(\mathcal{R}, \mathcal{G}, x) = \min_{j \in \mathcal{A}} \overline{c}_j^*(m_j(x), x) = \min_{j \in \mathcal{A}} \left\{ \overline{c}_0^*(x), \overline{c}_{j>0}(m_j(x), x) \right\}
\tag{38}
$$

We can write the calibration gap as $\Delta \mathcal{C}_{\widetilde{\ell}_{\mathrm{def}}}(r, g, x) := \mathcal{C}_{\widetilde{\ell}_{\mathrm{def}}}(r, g, x) - \mathcal{C}_{\widetilde{\ell}_{\mathrm{def}}}^B(\mathcal{R}, \mathcal{G}, x) \geq 0$, it follows:

$$
\begin{aligned}
\Delta \mathcal{C}_{\widetilde{\ell}_{\mathrm{def}}}(r, g, x) &= \mathcal{C}_{\widetilde{\ell}_{\mathrm{def}}}(r, g, x) - \min_{j \in \mathcal{A}} \overline{c}_j^*(m_j(x), x) \\
&= \underbrace{\mathcal{C}_{\widetilde{\ell}_{\mathrm{def}}}(r, g, x) - \min_{j \in \mathcal{A}} \overline{c}_j(g(x), m_j(x), x)}_{A} + \underbrace{\left( \min_{j \in \mathcal{A}} \overline{c}_j(g(x), m_j(x), x) - \min_{j \in \mathcal{A}} \overline{c}_j^*(m_j(x), x) \right)}_{B}
\end{aligned}
\tag{39}
$$

**Term $B$:**  Let's first focus on $B$. We can write the following inequality:

$$B = \min_{j \in \mathcal{A}} \overline{c}_j(g(x), m_j(x), x) - \min_{j \in \mathcal{A}} \overline{c}_j^*(m_j(x), x) \leq \overline{c}_0(g(x), x) - \overline{c}_0^*(x) \tag{40}$$

Indeed, we have the following relationship:

1. if $\overline{c}_0(g(x), x) < \min_{j \in [J]} \overline{c}_j(m_j(x), x) \implies B = \overline{c}_0(g(x), x) - \overline{c}_0^*(x)$

2. if $\overline{c}_0(g(x), x) > \min_{j \in [J]} \overline{c}_j(m_j(x), x)$ and $\overline{c}_0^*(x) \leq \min_{j \in [J]} \overline{c}_j(m_j(x), x)$
   $\implies B = \min_{j \in [J]} \overline{c}_j(m_j(x), x) - \overline{c}_0^*(x) \leq \overline{c}_0(g(x), x) - \overline{c}_0^*(x)$

**Term $A$:**  Then using the term $A$:

$$
\begin{aligned}
A &= \mathcal{C}_{\widetilde{\ell}_{\text{def}}}(r, g, x) - \min_{j \in \mathcal{A}} \overline{c}_j(m_j(x), x) \\
&= \mathcal{C}_{\widetilde{\ell}_{\text{def}}}(r, g, x) - \inf_{r \in \mathcal{R}} \mathcal{C}_{\widetilde{\ell}_{\text{def}}}(r, g, x) \\
&= \sum_{j=0}^{J} \left( \overline{\tau}_j(g(x), m(x), x) \widetilde{\ell}_{01}^j(r, x, j) \right) - \inf_{r \in \mathcal{R}} \sum_{j=0}^{J} \left( \overline{\tau}_j(g(x), m(x), x) \widetilde{\ell}_{01}^j(r, x, j) \right)
\end{aligned}
\tag{41}
$$

Now, we introduce a change of variables to define a probability distribution $p = (p_0, \cdots, p_j) \in \Delta^{|\mathcal{A}|}$, accounting for the fact that $\tau_j$ does not inherently represent probabilities. Consequently, for each $j \in \mathcal{A}$, we obtain the following expression:

$$p_j = \frac{\overline{\tau}_j(g(x), m(x), x)}{\sum_{j=0}^{J} \overline{\tau}_j(g(x), m(x), x)} = \frac{\overline{\tau}_j}{\|\tau\|_1} \quad (\text{for } \tau = \{\tau_j \geq 0\}_{j \in \mathcal{A}}) \tag{42}$$

We then, have:

$$A = \|\tau\|_1 \left( \sum_{j=0}^{J} \left( p_j \widetilde{\ell}_{01}^j(r, x, j) \right) - \inf_{r \in \mathcal{R}} \sum_{j=0}^{J} \left( p_j \widetilde{\ell}_{01}^j(r, x, j) \right) \right) \tag{43}$$

Then, using Lemma 5.9, it leads to:

$$
\begin{aligned}
A &\leq \|\tau\|_1 \Psi^u(1) \left[ \sum_{j=0}^{J} \left( p_j \widetilde{\Phi}_{01}^{\rho, u, j}(r, x, j) \right) - \inf_{r \in \mathcal{R}} \sum_{j=0}^{J} \left( p_j \widetilde{\Phi}_{01}^{\rho, u, j}(r, x, j) \right) \right] \\
&= \|\tau\|_1 \Psi^u(1) \frac{1}{\|\tau\|_1} \left[ \sum_{j=0}^{J} \left( \overline{\tau}_j(g(x), m(x), x) \widetilde{\Phi}_{01}^{\rho, u, j}(r, x, j) \right) - \inf_{r \in \mathcal{R}} \sum_{j=0}^{J} \left( \overline{\tau}_j(g(x), m(x), x) \widetilde{\Phi}_{01}^{\rho, u, j}(r, x, j) \right) \right] \\
&= \Psi^u(1) \left[ \sum_{j=0}^{J} \left( \overline{\tau}_j(g(x), m(x), x) \widetilde{\Phi}_{01}^{\rho, u, j}(r, x, j) \right) - \inf_{r \in \mathcal{R}} \sum_{j=0}^{J} \left( \overline{\tau}_j(g(x), m(x), x) \widetilde{\Phi}_{01}^{\rho, u, j}(r, x, j) \right) \right] \\
&= \Psi^u(1) \left[ \mathcal{C}_{\widetilde{\Phi}_{\text{def}}^{\rho, u}}(r, x) - \mathcal{C}_{\widetilde{\Phi}_{\text{def}}^{\rho, u}}^*(\mathcal{R}, x) \right]
\end{aligned}
\tag{44}
$$

Then, adding $B$ leads to:

$$
\begin{aligned}
\Delta \mathcal{C}_{\widetilde{\ell}_{\text{def}}}(r, g, x) &= A + B \quad (\text{using Eq. 39}) \\
&\leq \Psi^u(1) \left[ \mathcal{C}_{\widetilde{\Phi}_{\text{def}}^{\rho, u}}(r, x) - \mathcal{C}_{\widetilde{\Phi}_{\text{def}}^{\rho, u}}^*(\mathcal{R}, x) \right] + \overline{c}_0(g(x), x) - \overline{c}_0^*(x)
\end{aligned}
\tag{45}
$$

By construction, we have $\overline{c}_0(g(x), x) = \mathbb{E}_{y, t | x}[c_0(g(x), z)]$ with $c_0(g(x), z) = \psi(g(x), z)$ and $\overline{c}_0^*(x) = \inf_{g \in \mathcal{G}} \mathbb{E}_{y, t | x}[c_0(g(x), z)]$. Therefore, we can write for $g \in \mathcal{G}$ and the cost $c_0$:

$$\Delta \mathcal{C}_{c_0}(g, x) = \overline{c}_0(g(x), x) - \overline{c}_0^*(x) \tag{46}$$

Then,

$$\Delta\mathcal{C}_{\widetilde{\ell}_{\text{def}}}(r, g, x) \leq \Psi^u(1)\Big[\mathcal{C}_{\widetilde{\Phi}^{\rho,u}_{\text{def}}}(r, x) - \mathcal{C}^*_{\widetilde{\Phi}^{\rho,u}_{\text{def}}}(\mathcal{R}, x)\Big] + \Delta\mathcal{C}_{c_0}(g(x), x)$$
$$= \Psi^u(1)\Big[\Delta\mathcal{C}_{\widetilde{\Phi}^{\rho,u}_{\text{def}}}(r, x)\Big] + \Delta\mathcal{C}_{c_0}(g, x) \tag{47}$$

Therefore, by definition:

$$\begin{aligned}
\mathcal{E}_{\widetilde{\ell}_{\text{def}}}(r, g) - \mathcal{E}^*_{\widetilde{\ell}_{\text{def}}}(\mathcal{R}, \mathcal{G}) + \mathcal{U}_{\widetilde{\ell}_{\text{def}}}(\mathcal{R}, \mathcal{G}) &= \mathbb{E}_x[\Delta\mathcal{C}_{\widetilde{\ell}_{\text{def}}}(r, g, x)] \\
&\leq \Psi^u(1)\mathbb{E}_x\Big[\Delta\mathcal{C}_{\widetilde{\Phi}^{\rho,u}_{\text{def}}}(r, x)\Big] + \mathbb{E}_x\Big[\Delta\mathcal{C}_{c_0}(g(x), x)\Big] \\
&= \Psi^u(1)\Big(\mathcal{E}_{\widetilde{\Phi}^{\rho,u}_{\text{def}}}(r) - \mathcal{E}^*_{\widetilde{\Phi}^{\rho,u}_{\text{def}}}(\mathcal{R}) + \mathcal{U}_{\widetilde{\Phi}^{\rho,u}_{\text{def}}}(\mathcal{R})\Big) \\
&\quad + \mathbb{E}_x\Big[\Delta\mathcal{C}_{c_0}(g(x), x)\Big] \\
&= \Psi^u(1)\Big(\mathcal{E}_{\widetilde{\Phi}^{\rho,u}_{\text{def}}}(r) - \mathcal{E}^*_{\widetilde{\Phi}^{\rho,u}_{\text{def}}}(\mathcal{R}) + \mathcal{U}_{\widetilde{\Phi}^{\rho,u}_{\text{def}}}(\mathcal{R})\Big) \\
&\quad + \mathbb{E}_x[\bar{c}_0(g(x), x)] - \mathbb{E}_x[\bar{c}^*_0(x)] \\
&= \Psi^u(1)\Big(\mathcal{E}_{\widetilde{\Phi}^{\rho,u}_{\text{def}}}(r) - \mathcal{E}^*_{\widetilde{\Phi}^{\rho,u}_{\text{def}}}(\mathcal{R}) + \mathcal{U}_{\widetilde{\Phi}^{\rho,u}_{\text{def}}}(\mathcal{R})\Big) \\
&\quad + \Delta\mathcal{E}_{c_0}(g)
\end{aligned} \tag{48}$$

where $\Delta\mathcal{E}_{c_0}(g) = \mathcal{E}_{c_0}(g) - \mathcal{E}^B_{c_0}(\mathcal{G}) + \mathcal{U}_{c_0}(\mathcal{G})$.

In the special case of the log-softmax ($u = 1$), we have that $\Psi^u(1) = \log(2)$.

$\square$

# E. Experiments details

We present empirical results comparing the performance of state-of-the-art Two-Stage Learning-to-Defer frameworks (Mao et al., 2023a; 2024d; Montreuil et al., 2025c) with our robust SARD algorithm. To the best of our knowledge, this is the first study to address adversarial robustness within the context of Learning-to-Defer.

All baselines use the log-softmax surrogate for $\Phi_{01}$ with $\Psi^{u=1}(v) = \log(1 + v)$ and $\Psi_e(v) = \exp(-v)$. Adversarial attacks and supremum evaluations over the perturbation region $B_p(x, \gamma)$ are evaluated using Projected Gradient Descent (Madry et al., 2018). For each experiment, we report the mean and standard deviation over four independent trials to account for variability in results. Experiments are conducted on one NVIDIA H100 GPU. Additionally, we make our scripts publicly available.

### E.1. Multiclass Classification Task

**Experts:** We assigned categories to three distinct experts: expert $M_1$ is more likely to be correct on 58 categories, expert $M_2$ on 47 categories, and expert $M_3$ on 5 categories. To simulate a realistic scenario, we allow for overlapping expertise, meaning that for some $x \in \mathcal{X}$, multiple experts can provide correct predictions. On assigned categories, an expert has a probability $p = 0.94$ to be correct, while following a uniform probability if the category is not assigned.

Agent costs are defined as $c_0(h(x), y) = \ell_{01}(h(x), y)$ for the model and $c_{j>0}(m^h_j(x), y) = \ell_{01}(m^h_j(x), y)$, consistent with (Mozannar & Sontag, 2020; Mozannar et al., 2023; Verma et al., 2023; Cao et al., 2023; Mao et al., 2023a). We report respective accuracies of experts in Table 4.

**Model:** We train the classifier offline using a ResNet-4 architecture (He et al., 2015) for 100 epochs with the Adam optimizer (Kingma & Ba, 2015), a learning rate of 0.1, and a batch size of 64. The checkpoint corresponding to the lowest empirical risk on the validation set is selected. Corresponding performance is indicated in Table 4.

**Baseline (Mao et al., 2023a):** We train a rejector using a ResNet-4 (He et al., 2015) architecture for 500 epochs, a learning rate of 0.005, a cosine scheduler, Adam optimizer (Kingma & Ba, 2015), and a batch size of 2048. We report performance of the checkpoints corresponding to the lower empirical risk on the validation set.

|  | Model | Expert $M_1$ | Expert $M_2$ | Expert $M_3$ |
|---|---|---|---|---|
| Accuracy | 61.0 | 53.9 | 45.1 | 5.8 |

*Table 4.* Agent accuracies on the CIFAR-100 validation set. Since the training and validation sets are pre-determined in this dataset, the agents' knowledge remains fixed throughout the evaluation.

**SARD:** We train a rejector using the ResNet-4 architecture (He et al., 2015) for 1500 epochs with a learning rate of 0.005, a cosine scheduler, the Adam optimizer (Kingma & Ba, 2015) with L2 weight decay of $10^{-4}$ acting as regularizer, and a batch size of 2048. The hyperparameters are set to $\rho = 1$ and $\nu = 0.01$. The supremum component from the adversarial inputs is estimated using PGD40 (Madry et al., 2018) with $\epsilon = 8/255$, the $\ell_\infty$ norm, and a step size of $\epsilon/40$, following the approach in (Mao et al., 2023b; Awasthi et al., 2023).

| Baseline | Clean | Untarg. | Targ. Model | Targ. $M_1$ | Targ. $M_2$ | Targ. $M_3$ |
|---|---|---|---|---|---|---|
| Mao et al. (2023a) | $72.8 \pm 0.4$ | $17.2 \pm 0.2$ | $61.1 \pm 0.1$ | $54.4 \pm 0.1$ | $45.4 \pm 0.1$ | $13.4 \pm 0.1$ |
| Our | $67.0 \pm 0.4$ | $49.8 \pm 0.3$ | $64.8 \pm 0.2$ | $62.4 \pm 0.3$ | $62.1 \pm 0.2$ | $64.8 \pm 0.3$ |

*Table 5.* Comparison of accuracy results between the proposed SARD and the baseline (Mao et al., 2023a) on the CIFAR-100 validation set, including clean and adversarial scenarios.

### E.2. Regression Task

**Experts:** We train three experts offline with three layers MLPs (128, 64, 32), each specializing in a specific subset of the dataset based on a predefined localization criterion. The first expert $M_1$ trains on Southern California (latitude lower than 36), the second expert $M_2$ on Central California (latitude between 36 and 38.5), and the last in Northern California (otherwise) representing a smaller area. MLPs are trained using a ReLU, an Adam optimizer (Kingma & Ba, 2015), a learning rate of 0.001, and 500 epochs. Agent costs are defined as $c_0(f(x), t) = \text{RMSE}(g(x), t)$ for the model and $c_{j>0}(m_j^f(x), t) = \text{RMSE}(m_j(x), t)$, consistent with (Mao et al., 2024d). We report respective RMSE of experts in Table E.2.

**Model:** We train a regressor using a two-layer MLP with hidden dimensions (64, 32) on the full training set. The model uses ReLU activations, the Adam optimizer (Kingma & Ba, 2015), a learning rate of 0.001, and is trained for 500 epochs. We report performance of the checkpoints corresponding to the lower empirical risk on the validation set. The model performance is reported in Table E.2.

|  | Model | Expert $M_1$ | Expert $M_2$ | Expert $M_3$ |
|---|---|---|---|---|
| RMSE | $0.27 \pm .01$ | $1.23 \pm .02$ | $1.85 \pm .02$ | $0.91 \pm .01$ |

*Table 6.* Agent RMSE on the California Housing validation set (20% of the dataset).

**Baseline (Mao et al., 2024d):** We train a rejector using a MLP (8,16) for 100 epochs, a learning rate of 0.01, a cosine scheduler, Adam optimizer (Kingma & Ba, 2015), and a batch size of 8096. We report performance of the checkpoints corresponding to the lower empirical risk on the validation set.

**SARD:** We train a rejector using a MLP (8,16) for 400 epochs, a learning rate of 0.01, a cosine scheduler, Adam optimizer (Kingma & Ba, 2015) with L2 weight decay of $10^{-4}$ acting as regularizer, and a batch size of 8096. The hyperparameters are set to $\rho = 1$ and $\nu = 0.05$. The supremum component from the adversarial inputs is estimated using PGD10 (Madry et al., 2018) with $\epsilon$ equal to 25% of the variance of dataset's features, the $\ell_\infty$ norm, and a step size of $\epsilon/10$, following the approach in (Mao et al., 2023b; Awasthi et al., 2023).

| Baseline | Clean | Untarg. | Targ. Model | Targ. $M_1$ | Targ. $M_2$ | Targ. $M_3$ |
|---|---|---|---|---|---|---|
| Mao et al. (2024d) | $0.17 \pm 0.01$ | $0.29 \pm 0.3$ | $0.19 \pm 0.01$ | $0.40 \pm 0.02$ | $0.21 \pm 0.01$ | $0.41 \pm 0.05$ |
| Our | $0.17 \pm 0.01$ | $0.17 \pm 0.01$ | $0.17 \pm 0.01$ | $0.18 \pm 0.01$ | $0.18 \pm 0.01$ | $0.18 \pm 0.01$ |

*Table 7.* Performance comparison of SARD with the baseline (Mao et al., 2024d) on the California Housing dataset. The table reports Root Mean Square Error (RMSE) under clean and adversarial scenarios.

### E.3. Multi Task

**Experts:** We train two specialized experts using a Faster R-CNN (Ren et al., 2015) architecture with a MobileNet (Howard et al., 2017) backbone. The first expert, $M_1$, is trained on images containing *animals*, while the second expert, $M_2$, is trained on images containing *vehicles*. Both experts are trained using the Adam optimizer (Kingma & Ba, 2015) with a learning rate of 0.005, a batch size of 128, and trained for 50 epochs. Agent costs are defined as $c_0(g(x), z) = \text{mAP}(g(x), z)$ for the model and $c_{j>0}(m_j(x), z) = \text{mAP}(m_j(x), z)$, consistent with (Montreuil et al., 2025c). We report respective mAP of experts in Table E.3.

**Model:** We train an object detection model using a larger Faster R-CNN (Ren et al., 2015) with ResNet-50 FPN (He et al., 2015) backbone. We train this model with Adam optimizer (Kingma & Ba, 2015), a learning rate of 0.005, a batch size of 128, and trained for 50 epochs. We report performance of the checkpoints corresponding to the lower empirical risk on the validation set. The model performance is reported in Table E.3.

| | Model | Expert $M_1$ | Expert $M_2$ |
|---|---|---|---|
| mAP | 39.5 | 17.2 | 20.0 |

*Table 8.* Agents mAP Pascal VOC validation set. Since the training and validation sets are pre-determined in this dataset, the agents' knowledge remains fixed throughout the evaluation.

**Baseline (Montreuil et al., 2025c):** We train a rejector using a Faster R-CNN (Ren et al., 2015) with a MobileNet backbone (Howard et al., 2017) and a classification head. We train this rejector for 70 epochs, a learning rate $5e^{-4}$, a cosine scheduler, Adam optimizer (Kingma & Ba, 2015), and a batch size of 256. We report performance of the checkpoints corresponding to the lower empirical risk on the validation set.

**SARD:** We train a rejector using a Faster R-CNN (Ren et al., 2015) with a MobileNet backbone (Howard et al., 2017) and a classification head for 70 epochs with a learning rate of 0.001, a cosine scheduler, the Adam optimizer (Kingma & Ba, 2015) with L2 weight decay of $10^{-4}$ acting as regularizer, and a batch size of 64. The hyperparameters are set to $\rho = 1$ and $\nu = 0.01$. The supremum component from the adversarial inputs is estimated using PGD20 (Madry et al., 2018) with $\epsilon = 8/255$, the $\ell_\infty$ norm, and a step size of $\epsilon/20$, following the approach in (Mao et al., 2023b; Awasthi et al., 2023).

| Baseline | Clean | Untarg. | Targ. Model | Targ. $M_1$ | Targ. $M_2$ |
|---|---|---|---|---|---|
| Montreuil et al. (2025c) | $44.4 \pm 0.4$ | $9.7 \pm 0.1$ | $39.5 \pm 0.1$ | $17.4 \pm 0.2$ | $20.4 \pm 0.2$ |
| Our | $43.9 \pm 0.4$ | $39.0 \pm 0.3$ | $39.5 \pm 0.1$ | $39.7 \pm 0.3$ | $39.6 \pm 0.1$ |

*Table 9.* Performance comparison of SARD with the baseline (Montreuil et al., 2025c) on the Pascal VOC dataset. The table reports mean Average Precision (mAP) under clean and adversarial scenarios.

