# OpenReview forum: "Adversarial Robustness in Two-Stage Learning-to-Defer: Algorithms and Guarantees"
_ICML.cc/2025/Conference — ICML 2025 poster_

### Official Review · Reviewer_DEy6 · 2025-02-20

**Overall Recommendation:** 4

**Summary:**

This paper studies adversarial robustness in the L2D paradigm. Based on rigorous theoretical results, they propose a novel method called SARD to improve the adversarial robustness of L2D models.

Concretely, this paper first presents untargeted and targeted attacks on the L2D, based on optimization on the commonly adopted adversarial loss, tailored to the L2D setting. They then develop approximations of the worst-case loss and then apply smoothing (lemma 5.3) on the proposed loss to ease optimization. Consistency bounds are then developed for the losses.

**Claims And Evidence:**

The claims are sufficiently supported.

Concretely, the authors claim the development of novel attack and adversarial training methods on the L2D, and claim consistency bounds on the correponsding loss functions.

**Essential References Not Discussed:**

The literature is sufficiently discussed.

**Experimental Designs Or Analyses:**

The authors checked the performance of the proposed methods in classification, regression, and multi-task settings, which sufficiently supported the generalization of their methods. However, it would be better if they could evaluate more datasets for each setting; currently, each task involves a single dataset.

**Methods And Evaluation Criteria:**

The method makes sense, and the evaluation has no evident flaw.

**Other Comments Or Suggestions:**

None.

**Other Strengths And Weaknesses:**

Since this work is the first to discuss adversarial robustness for a specific family of models (L2D), I recommend accept.

**Questions For Authors:**

Could you include more datasets in the evaluation of each task?

**Relation To Broader Scientific Literature:**

To the best of my knowledge, this paper is the first to evaluate adversarial robustness and effectively propose an adversarial training method for the L2D paradigm.

While the authors do not present many reasoning about why this is important, I believe that evaluating from adversarial perspective for a possibly deployed paradigm is useful.

This work is not of general interest to the adversarial machine learning community because it only studies a specific setting, but it is of interest to those studying L2D.

**Theoretical Claims:**

I did not check the formal proofs; nevertheless, the derived method based on their theories work quite well.

---

> ### Author Rebuttal · Authors · 2025-03-29
>
> We sincerely thank the reviewer for their encouraging and constructive feedback. We appreciate their recognition of the rigor of our theoretical contributions and the strength of our experimental results in supporting our claims. We also acknowledge and value their observation that, to the best of their knowledge, this is the first work to study adversarial robustness in L2D. Please find our clarifications below.
>
> > The authors checked the performance of the proposed methods in classification, regression, and multi-task settings, which sufficiently supported the generalization of their methods. However, it would be better if they could evaluate more datasets for each setting; currently, each task involves a single dataset.
>
> > Could you include more datasets in the evaluation of each task?
>
> We would like to emphasize that the primary objective of our paper is to establish a rigorous theoretical foundation for defending against adversarial attacks in the Learning-to-Defer  setting. Our experiments primarily serve to empirically validate the proposed theoretical framework, rather than to benchmark performance across a wide range of datasets. Given the theoretical focus and current standards within the L2D community, we believe our existing experiments across classification, regression, and multi-task scenarios sufficiently demonstrate generality (see discussion with reviewer @dgrk).
>
> Nonetheless, exploring more datasets remains a valuable direction for future empirical work.
>
> > While the authors do not present many reasoning about why this is important, I believe that evaluating from adversarial perspective for a possibly deployed paradigm is useful.
>
> Thank you for recognizing the importance of evaluating Learning-to-Defer from an adversarial perspective. We strongly agree that such robustness evaluation is crucial. **L2D models are increasingly deployed in high-stakes applications, including healthcare [10, 11], or more broadly autonomous decision-making, where adversarial manipulation leading to incorrect deferral decisions can have severe real-world consequences**. Specifically, targeted attacks can strategically force the rejector to allocate more queries to a particular agent, thereby causing a harmful bias in the allocation process. On the other hand, untargeted attacks degrade the overall performance and reliability of the entire system, making its behavior unpredictable by maximizing the occurrence of errors.
>
> We further discuss potential misuse of L2D in the discussion with reviewer @689i and will clarify these critical points in our revised manuscript.
>
>
> ### References
> [10] Strong et al. (2025). Towards Human-AI Collaboration in Healthcare: Guided Deferral Systems with Large Language Models. AAAI25
>
> [11] Joshi et al. (2021). Learning-to-defer for sequential medical decision-making under uncertainty. TMLR21

---

> > ### Comment · Reviewer_DEy6 · 2025-04-02
> >
> > Dear authors,
> >
> > Thanks for the rebuttal. This sufficiently addresses my questions.

---

### Official Review · Reviewer_BFHK · 2025-03-11

**Overall Recommendation:** 3

**Summary:**

This paper addresses adversarial robustness in two-stage Learning-to-Defer (L2D) frameworks by introducing two new attack strategies: untargeted attacks, which disrupt agent allocation, and targeted attacks, which redirect queries to specific agents. To counter these attacks, the authors propose SARD, a robust, convex deferral algorithm grounded in Bayes and (R, G)-consistency. Experimental results validate both the effectiveness of the attacks and the robustness of the proposed defense.

**Claims And Evidence:**

Yes.

**Essential References Not Discussed:**

Yes.

**Experimental Designs Or Analyses:**

The experimental results seem to be sound.

**Methods And Evaluation Criteria:**

Yes.

**Other Comments Or Suggestions:**

N/A.

**Other Strengths And Weaknesses:**

- One issue with the paper is that Section 3 (Preliminaries) begins with the multi-task scenario, which may misleadingly suggest that the paper primarily focuses on multi-task learning. This raises questions: Is Learning-to-Defer (L2D) predominantly studied in the multi-task setting, or is adversarial robustness particularly relevant to multi-task learning? If neither is true, the choice to emphasize multi-task learning from the outset needs clearer justification. A possible improvement would be to restructure the section by first introducing classification, then regression, and finally discussing multi-task learning as an extension.

- The paper does not sufficiently highlight the challenges and technical difficulty of the problem. One of the main contributions—introducing two new attack strategies—feels somewhat straightforward, as it mainly exploits existing vulnerabilities rather than proposing novel attack methods. Similarly, while the proposed defense incorporates consistency guarantees, it appears to be largely a combination of existing theories from learning-to-defer consistency and H-consistency bounds for adversarial robustness. The technical difficulty of deriving SARD beyond this theoretical stacking is unclear.

**Questions For Authors:**

N/A.

**Relation To Broader Scientific Literature:**

The results of this paper are related to the literature of learning-to-defer and adversarial robustness.

**Theoretical Claims:**

Theoretical claims look sound.

---

> ### Author Rebuttal · Authors · 2025-03-29
>
> We thank the reviewer for their thoughtful evaluation and appreciate the recognition of our experiments and theoretical claims. In the following, we clarify our motivations, provide justification for our design choices (e.g., the multi-task emphasis in Section 3), and highlight the technical challenges addressed in deriving SARD. We also discuss how our proposed attacks go beyond existing methods and why they are theoretically and practically significant in the L2D context.
> > One issue with the paper is that Section 3 begins with the multi-task [...] discussing multi-task learning as an extension.
>
> Our motivation for initially framing Section 3 within the multi-task setting was not to imply that L2D is predominantly studied or especially relevant only to multi-task problems. Rather, we aimed to emphasize the generality of our theoretical results, demonstrating explicitly that the agent costs $c_j$ can take any positive form—classification [5], regression [6], or any multi-task metrics. **We intended to highlight that our proofs, losses, attacks, and defense methods hold broadly across these different problem formulations, a fact that is non-trivial [5,6] and significant for demonstrating the versatility of our approach**.
>
> In simple words, our approach (SARD) can be used in any setting. We thank you for pointing out this potential misunderstanding and will clarify this in the final version.
> > The paper does not sufficiently highlight the challenges and technical difficulty of the problem [...] The technical difficulty of deriving SARD beyond this theoretical stacking is unclear.
>
> While the base attack strategies we adapt were initially developed for multiclass classification, a key contribution of our work lies in their novel extension to the fundamentally different setting of L2D. **Specifically, we are the first to show explicitly how these attack methods can be repurposed to strategically corrupt query routing decisions, which we believe is not trivial**. Importantly, our attacks target the rejector itself (responsible for allocation decisions), rather than the agents performing the tasks, which is an important distinction to multiclass classification. This design choice stems from a core characteristic of the two-stage L2D setting: we do not have access to the internal structure or training procedure of the agents, as also assumed in [4,5].
>
> We provide a concrete example and detailed discussion (see response to reviewer @689i) highlighting why robustness against such novel attacks is critical, given the high-stakes decisions typically addressed by L2D systems.
>
> Regarding technical complexity, we emphasize that the losses, attack strategies, and theoretical results introduced in our paper are entirely novel and require dedicated analysis, as formalized in Theorem 5.7. While previous works (e.g., [7,8,9]) have studied adversarial surrogate losses, their analyses are confined to multiclass classification and do not extend naturally to the L2D. **As discussed with reviewer @689i, applying standard adversarial training directly to the L2D loss (Definition 3.1) overlooks the worst-case allocation scenario—captured in Lemma 5.1—and thus leaves the system vulnerable to attacks**. Moreover, even in the absence of adversarial perturbations, proving consistency guarantees for L2D is nontrivial, as demonstrated in recent literature [3,4,5,6,12]. Furthermore, Lemma 5.6 and its proof are entirely original and differ from prior results (e.g., [7,8,9]). Notably, our theoretical framework explicitly addresses worst-case deferral loss by introducing adversarial inputs tailored for each individual agent $j \in \mathcal{A}$—a novel and technically challenging aspect that is not present in standard multiclass classification analyses.
>
> We hope this clarifies the challenges involved, and we would be glad to elaborate further if needed.
>
> ### References
>
> [3] Mozannar et al. (2021). Consistent estimators for learning to defer to an expert. ICML21
>
> [4] Verma et al. (2023). Learning to Defer to Multiple Experts: Consistent Surrogate Losses, Confidence Calibration, and Conformal Ensembles. AISTATS23
>
> [5] Mao, et al. (2023). Two-Stage Learning to Defer with Multiple Experts. NeurIPS23
>
> [6] Mao, et al. (2024). Regression with multi-expert deferral. NeurIPS24
>
> [7] Awasthi, et al. (2023). Theoretically Grounded Loss Functions and Algorithms for Adversarial Robustness. AISTATS23
>
> [8] Mao et al. (2023). Cross-entropy loss functions: theoretical analysis and applications. ICML23
>
> [9] Bao, et al. (2021) Calibrated surrogate
> losses for adversarially robust classification. COLT21
>
> [10] Strong et al. (2025). Towards Human-AI Collaboration in Healthcare: Guided Deferral Systems with Large Language Models. AAAI25
>
> [11] Joshi et al. (2021). Learning-to-defer for sequential medical decision-making under uncertainty. TMLR21
>
> [12] Mao et al. (2024). Principled Approaches for Learning to Defer with Multiple Experts. ISAIM24

---

### Official Review · Reviewer_689i · 2025-03-14

**Overall Recommendation:** 3

**Summary:**

This paper identifies that Learning-to-Defer frameworks are vulnerable to adversarial attacks and introduces two attack strategies: untargeted attacks that disrupt allocation and targeted attacks that redirect queries to specific agents. The authors propose SARD, a robust algorithm with theoretical guarantees based on Bayes-consistency and (R,G)-consistency. Experiments that while existing frameworks suffer severe performance degradation under attacks, SARD maintains consistent performance in both clean and adversarial conditions.

**Claims And Evidence:**

The main claims are well-supported by evidence。 The vulnerability of existing L2D frameworks is convincingly demonstrated through empirical results. And the robustness of SARD is supported by consistent performance across clean and adversarial conditions in all three tasks.

**Essential References Not Discussed:**

It covers most essential reference

**Experimental Designs Or Analyses:**

The experiments overall validate the theoretical claims and demonstrate the practical benefits of SARD across different application domains.

**Methods And Evaluation Criteria:**

The methods and evaluation criteria are appropriate for the problem with diverse tasks and sufficiently large dataset.

**Other Comments Or Suggestions:**

- There is a noticeable performance trade-off - SARD consistently achieves slightly lower performance on clean data compared to baselines in exchange for robustness. While this is a common challenge in adversarial robustness research, a more explicit discussion of this trade-off would be valuable.
- There is a noticeable performance trade-off - SARD consistently achieves slightly lower performance on clean data compared to baselines in exchange for robustness. While this is a common challenge in adversarial robustness research, a more explicit discussion of this trade-off would be valuable.
- Minor typo on page 5: "we define the j-th adversarial true multiclass loss" appears to be missing the complete definition.

**Other Strengths And Weaknesses:**

- The computational complexity and training overhead of SARD compared to baseline methods are not discussed, which is important for practical deployment considerations.
- The paper doesn't explore whether standard adversarial training approaches could be directly applied to the baseline models as an alternative solution.

**Questions For Authors:**

- How does SARD's performance change as the perturbation budget increases? Is there a point at which the theoretical guarantees break down, and if so, how does this compare to standard adversarial training approaches?
- Did you explore the effectiveness of standard adversarial training approaches (e.g., PGD-AT) applied directly to the baseline models? This would help clarify whether the benefits come specifically from your novel formulation or could be achieved with simpler adaptations of existing methods.
- How sensitive is SARD to the choice of hyperparameters ρ and ν? Did you observe any consistent patterns during hyperparameter tuning that could serve as practical guidelines for implementation?
- For real-world deployment in critical applications, how would you recommend practitioners balance the trade-off between clean performance and robustness? Are there specific scenarios where you believe the robustness benefits would clearly outweigh the slight decrease in clean performance?

**Relation To Broader Scientific Literature:**

It provides an valueable perspective.

**Theoretical Claims:**

The theoretical claims are sound and well-supported by detailed proofs:

Lemma 5.6 establishes R-consistency bounds for the j-th adversarial margin surrogate losses, building on consistency theory for adversarially robust classification.
Theorem 5.7 extends these bounds to the full adversarial margin deferral surrogate losses, showing how they relate to the adversarial true deferral loss.

---

> ### Author Rebuttal · Authors · 2025-03-29
>
> We sincerely thank the reviewer for their thoughtful and constructive feedback. We are grateful for your recognition of the rigor in our theoretical contributions and the strength of our empirical validation.
>
> > The computational complexity [...] practical deployment considerations.
>
> Thank you for highlighting this important consideration. Let $\mathcal{F}$ denote the computational cost of performing a single forward–backward pass for the rejector model $r \in \mathcal{R}$. For standard L2D approaches involving $|\mathcal{A}| = J + 1$ agents, the complexity is $\mathcal{O}(\mathcal{F} + J)$. SARD, involves performing adversarial training through a PGD, which requires $T$ forward–backward passes per agent. Consequently, the computational complexity of SARD is $\mathcal{O}\bigl((J+1) T \mathcal{F}\bigr)$. We will explicitly clarify this complexity comparison in the revised manuscript.
>
> > The paper doesn't explore whether [...] as an alternative solution.
>
> **We demonstrate that naively applying standard adversarial training to existing L2D baselines (Definition 3.1) fails to minimize the desired worst-case deferral loss (as shown in Lemma 5.1)**. As a result, an attacker could still exploit the allocation mechanism if adversarial training is applied to the non-worst-case deferral loss defined in Definition 3.1.
>
> This motivates our design and optimization of a distinct loss function—one that fundamentally departs from the standard adversarial training objective. We will further clarify this in the main body of the paper.
>
> > Minor typo on page 5: [...] missing the complete definition.
>
> Thank you for pointing this out. We will correct this.
>
> > How does SARD's performance change [...] compare to standard adversarial training approaches?
>
> We did not observe any unexpected behavior from SARD compared to standard adversarial training [7,8,9]. Intuitively, as the adversarial budget increases, the problem becomes inherently more challenging to defend against. From a theoretical standpoint, SARD does not exhibit any particular breakdowns beyond those already known for standard adversarial training.
>
> > Did you explore the effectiveness [...] with simpler adaptations of existing methods.
>
> Yes, we explored this explicitly in our experiments. In each considered scenario, we evaluated baseline approaches both on clean datasets and under our novel attacks with a PGD [13]. **We demonstrate that existing baseline methods are highly vulnerable to both our attacks, while our proposed approach, SARD, consistently outperforms these baselines by a significant margin under adversarial conditions**.
>
> > How sensitive is SARD [...] practical guidelines for implementation?
>
> Indeed, SARD's performance depends on these hyperparameters, similar to smoothing approach in adversarial training [7,8]. Unlike previous observations in multiclass classification [7, 8], SARD generally achieves better performance with relatively smaller values of $\nu$. This suggests that our L2D formulation inherently requires a lighter adversarial regularization term, likely due to the increased complexity of the decision boundaries involved.
>
> We will explicitly clarify these practical guidelines.
>
> > For real-world deployment [...] benefits would clearly outweigh the slight decrease in clean performance?
>
> **L2D frameworks are specifically designed for scenarios where decision-making carries important risks, thereby making robustness an essential consideration**. Consider a hospital deploying L2D for cancer diagnosis. Ideally, L2D would allocate cancer detection queries to the most suitable agent available. Suppose the hospital has several agents: a neurologist, a dermatologist, a general practitioner, and a properly trained AI model, each with distinct consultation costs (e.g., neurologist: $5\beta_1$, dermatologist: $5\beta_1$, general practitioner: $\beta_1$, AI model: $0$).
>
> In this realistic scenario, an adversary might deliberately manipulate the L2D system (rejector) through the attacks we have introduced: our untargeted attack (Definition 4.1) could cause misallocation of queries, leading to critical cases, such as complex skin cancer detection, being incorrectly routed from the dermatologist to a less specialized agent like the AI model, increasing the risk of making a mistake. **Alternatively, using our targeted attack (Definition 4.2), an adversary might intentionally route straightforward cases to expensive experts (e.g., neurologist) to unnecessarily increase costs ($5\beta_1$ instead of $0$), or maliciously redirect consultations to a collaborating agent motivated by financial gain**.
>
> Given these potential vulnerabilities, we firmly believe that the benefits of robustness clearly outweigh minor decreases in clean-data performance in such high-stakes settings. This motivates our approach, especially as L2D frameworks are increasingly adopted in critical applications [10, 11].
>
> See discussion with reviewer @BFHK for references.

---

### Official Review · Reviewer_dgrk · 2025-03-18

**Overall Recommendation:** 3

**Summary:**

This paper investigates the two-stage learning to defer (L2D) frameworks under adversarial attacks. The authors introduce two novel attacks: untargeted and targeted, that exploit structural weaknesses in L2D systems. Then the authors propose the SARD algorithm, a robust, convex deferral mechanism that is both Bayes and (R,G)-consistency. SARD ensures optimal task allocation under adversarial perturbations and demonstrates robust performance across classification, regression, and multi-task benchmarks.

**Claims And Evidence:**

All the theoretical claims are supported by detailed proofs.

**Essential References Not Discussed:**

1. (1) and the j-th adversarial margin surrogate closely resembles the structure of Gamma-Phi losses, whose consistency and construction are thoroughly analyzed in [1]. A detailed discussion of this work is encouraged, examining whether the conclusions in [1] can simplify the proofs in this paper or inspire new insights relevant to this study.

2. The training of the allocation rule follows a post-hoc approach, similar to the framework of the post-hoc estimator for L2D [2].

[1]. Wang, Y. and Scott, C. On classification-calibration of gamma-phi losses. In Conference on Learning Theory, pages 4929–4951, 2023.

[2]. Narasimhan, H., Jitkrittum, W., Menon, A., Rawat, A., Kumar, S. Post-hoc estimators for learning to defer to an expert. Advances in Neural Information Processing Systems, 2022.

**Experimental Designs Or Analyses:**

In the experiments, various type of expert predictions are simulated for different datasets, while the soundness of the experimental design can be improved if results on datasets with real-world expert predictions (e.g., CIFAR-10H) can be included.

**Methods And Evaluation Criteria:**

The evaluation criteria and selected baseline methods are appropriate for assessing the two types of attacks.

**Other Comments Or Suggestions:**

There are some minor formatting issues in the citations. For example, citations [1] and [2] should refer to their published versions rather than preprints.

[1]. He, K., Zhang, X.,Ren, S., and Sun, J. Deep residual learning for image recognition, 2015. URLhttps://arxiv.org/abs/1512.03385.

[2]. Mao, A., Mohri, M., and Zhong, Y. Realizable h-consistent and bayes-consistent loss functions for learning to defer, 2024c. URLhttps://arxiv.org/abs/2407.13732.

**Other Strengths And Weaknesses:**

1. The setting of this paper is comprehensive, encompassing classification, regression, and their multitask variants. By addressing multiple learning paradigms, the study ensures broad applicability and relevance across diverse machine learning tasks.

2. The proposed attacks are intuitive and align naturally with real-world adversarial scenarios.

**Questions For Authors:**

This paper primarily focuses on attacks targeting the deferral rule, whereas attacks on the classification rule are more common in the multiclass classification setting. In general, these two types of attacks are not mutually exclusive. It is recommended to explore their combination as a direction for future work.

**Relation To Broader Scientific Literature:**

While the authors focuses on the problem of adversarial robustness, the authors further discuss the H-consistency of the proposed methods.

**Theoretical Claims:**

I reviewed the proof of Lemma 5.1 and did not find any apparent flaws.

---

> ### Author Rebuttal · Authors · 2025-03-29
>
> We thank the reviewer for their thoughtful and constructive feedback. We are glad that they
> found our contributions meaningful, and we appreciate their recognition of the soundness
> of both our theoretical and empirical results.
> > In the experiments, [...] with real-world expert predictions (e.g., CIFAR-10H) can be included.
>
> **We want to emphasize that the primary contribution of our paper is theoretical**. The experiments presented are primarily intended to empirically support and illustrate these theoretical results.
>
> Within the L2D community, synthetic experts are widely preferred for theoretical evaluation precisely because they allow controlled, systematic analyses across diverse conditions (e.g., specialized expert behaviors, and critical edge-case scenarios) that are generally not reproducible with currently available real-world expert labels [2,3,4,5,6]. For instance, we introduce synthetic experts to rigorously expose vulnerabilities to our novel targeted attack—a scenario that is difficult to construct using existing real-world dataset, yet remains plausible in practical deployments.
>
> Nonetheless, we acknowledge the value of evaluating real-world datasets and consider this for future empirical exploration.
> > j-th adversarial margin surrogate closely resembles the structure of Gamma-Phi losses [...] relevant to this study.
>
> Very good question! We confirm that our surrogate can be rewritten in a Gamma-Phi form $\widetilde{\Phi}^{\rho,u,j}_{01}(r, x, j)= \sup\_{x\_j' \in B\_p} \gamma ( \sum\_{j' \neq j} \phi \big( r(x\_j', j') - r(x\_j', j) \big))$ with $\gamma(v)=\log(1+v)$ (assuming $v=1$) and $\phi(v)=\min(\max(0,1 - v/\rho), 1)$. However, despite satisfying the Gamma-PD condition (Definition 3.1 in [1]), our surrogate does not satisfy Definition 3.2 (Phi-NDZ), as $\phi(v)$ is not differentiable on $\mathbb{R}$. Thus, we cannot directly leverage the conclusions of Theorem 2.6 from [1] to prove classification-calibration.
>
> Furthermore, the classification-calibration provided by [1] implicitly assumes the hypothesis class includes all measurable functions $\mathcal{R}_{\text{all}}$, a strong assumption that does not hold in our analysis. **Additionally, while classification-calibration implies Bayes-consistency—an asymptotic property—our Theorem 5.7 provides stronger finite-sample guarantees via explicit inequalities, without highly restricting the hypothesis class $\mathcal{R}$**.
>
> We will explicitly discuss these distinctions in the revised manuscript.
> > The training [...] post-hoc estimator for L2D [2].
>
> Thanks for the suggestion—we will include it. In fact, [5] can be viewed as an extension to the multi-expert setting, whereas [2] addresses only the single-expert case.
> > There are some minor formatting issues in the citations [...] preprints.
>
> Thank you for pointing this out.  We will correct this in the revised manuscript
> > This paper primarily focuses on attacks targeting the deferral rule, whereas attacks on the classification rule  [...] combination as a direction for future work.
>
> We agree that investigating combined attacks represents an interesting direction for future research.
>
> However, we would like to emphasize that the motivation for our work stems from the two-stage L2D setting [2,5,6], where the rejector is solely responsible for allocating each query to an external agent. In this setup, the external agents are fixed and not accessible beyond their output predictions. That is, we do not have access to their internal parameters, decision boundaries, or training pipelines; moreover, they may not even be describable as functions (e.g. human decision-makers).
>
> As such, while we acknowledge that attacks on the classification and deferral rules are not mutually exclusive in general, the structure of our problem precludes joint attacks. We cannot meaningfully design or evaluate perturbations that target the expert prediction, as we lack the ability to interact with or analyze the internal behavior of the experts. This naturally restricts our adversarial analysis to the deferral mechanism, which is the only component under the learner’s control.
>
> **Moreover, robustness in classification has been extensively studied. In contrast, robustness in L2D systems remains unexplored. Our work aims to address this gap**.
>
> We will clarify this in the revised manuscript.
> ### References
> [1] Wang et al. (2023). On classification-calibration of gamma-phi losses. COLT23
>
> [2] Narasimhan et al. (2022) Post-hoc estimators for learning to defer to an expert. NeurIPS22
>
> [3] Mozannar et al. (2021). Consistent estimators for learning to defer to an expert. ICML21
>
> [4] Verma et al. (2023). Learning to Defer to Multiple Experts: Consistent Surrogate Losses, Confidence Calibration, and Conformal Ensembles. AISTATS23
>
> [5] Mao, et al. (2023). Two-Stage Learning to Defer with Multiple Experts. NeurIPS23
>
> [6] Mao, et al. (2024). Regression with multi-expert deferral. NeurIPS24

---

### Decision · Program_Chairs · 2025-05-01

**Decision:**

Accept (poster)

**Comment:**

In the Learning to Defer (L2D) paradigm, a classifier is allowed to defer to an expert on samples it deems to be of low confidence. This paper studies the vulnerability of two-stage L2D approaches to adversarial attacks, and proposes a family of surrogate losses for L2D that are robust to such attacks. They establish consistency guarantees for the proposed losses and demonstrate their efficacy through experiments.

The reviewers are generally positive about the paper, and the authors satisfactorily addressed the questions raised during the rebuttal process.  Aside from concerns that the paper addresses a niche area, and requires a trade-off between performance and robustness, the reviewers were appreciative of the theoretical contributions.

We are happy to accept this paper, and strongly encourage the authors to incorporate the promised clarifications in their response, including discussions on the computational complexity, on the technical challenges involved, on additional related papers, and on practical use-cases in high-stakes application.